# Particle exchange statistics beyond fermions and bosons

Zhiyuan Wang[1,2,3 ✉] & Kaden R. A. Hazzard[1,2]

It is commonly believed that there are only two types of particle exchange statistics in quantum mechanics, fermions and bosons, with the exception of anyons in two dimensions[1–5]. In principle, a second exception known as parastatistics, which extends outside two dimensions, has been considered[6] but was believed to be physically equivalent to fermions and bosons[7–9]. Here we show that non-trivial parastatistics inequivalent to either fermions or bosons can exist in physical systems. These new types of identical particle obey generalized exclusion principles, leading to exotic free-particle thermodynamics distinct from any system of free fermions and bosons. We formulate our theory by developing a second quantization of paraparticles that naturally includes exactly solvable non-interacting theories and incorporates physical constraints such as locality. We then construct a family of exactly solvable quantum spin models in one and two dimensions, in which free paraparticles emerge as quasiparticle excitations, and their exchange statistics can be physically observed and are notably distinct from fermions and bosons. This demonstrates the possibility of a new type of quasiparticle in condensed matter systems and—more speculatively—the potential for previously unconsidered types of elementary particle.

It is commonly believed that there are only two types of particle exchange statistics—fermions and bosons. The standard textbook argument for this dichotomy is as follows. Each multiparticle quantum state is described by a wavefunction $\Psi(x_1, x_2,..., x_n)$, a complex-valued function of particle coordinates in a $d$-dimensional space $x_1, x_2, ..., x_n \in \mathbb{R}^d$. The particles are identical, meaning that, when we exchange any two of them (say, $x_1$ and $x_2$), the resulting wavefunction $\Psi(x_2, x_1,..., x_n)$ must represent the same physical state and therefore can change by at most a constant factor

$$\Psi(x_2, x_1, ..., x_n) = c\Psi(x_1, x_2, ..., x_n). \tag{1}$$

If we perform a second exchange, we have

$$\begin{aligned} \Psi(x_1, x_2, ..., x_n) &= c\Psi(x_2, x_1, ..., x_n) \\ &= c^2\Psi(x_1, x_2, ..., x_n), \end{aligned} \tag{2}$$

leading to $c^2 = 1$, as the wavefunction cannot be constantly zero. This provides exactly two possibilities, bosons ($c = 1$) and fermions ($c = -1$).

Despite being simple and convincing, there are two important exceptions to the fermion/boson dichotomy. The first is anyons in two spatial dimensions (2D)[1–5,10]. The second is parastatistics[6,11–15], which can be consistently defined in any dimension. The way this evades the above argument is that the wavefunction can carry extra indices that transform non-trivially during an exchange. Consider an $n$-particle wavefunction $\Psi(x_1, x_2,..., x_n)$, in which $I$ is a collection of extra indices corresponding to some internal degrees of freedom inaccessible to local measurements. Under an exchange between particles $j$ and $j + 1$

(note that we only need to specify the behaviour of the wavefunction under exchange of particles with adjacent labels, as non-adjacent exchanges can always be decomposed into a sequence of adjacent exchanges), the wavefunction may undergo a matrix transformation

$$\Psi^I(\{x_i\}_{i=1}^n)|_{x_j \leftrightarrow x_{j+1}} = \sum_J (R_j)^I_J \Psi^J(\{x_i\}_{i=1}^n), \tag{3}$$

for $j = 1,..., n - 1$, in which the summation is over all possible values of $J$. Similar to the $c^2 = 1$ constraint for equation (1), the matrices $(R_j)^I_J$ have to satisfy some algebraic constraints to guarantee consistency of equation (3):

$$\tag{4}$$

$$R_j^2 = 1, \qquad R_{j-1} R_j R_{j-1} = R_j R_{j-1} R_j$$

and $R_i R_j = R_j R_i$ for $|i - j| \geq 2$. The derivation of the first equation is similar to equation (2), the second equation is because of the equivalence of two different ways of swapping $x_{j-1}, x_j, x_{j+1}$ to $x_{j+1}, x_j, x_{j-1}$ and the last equation is because of the commutativity of the swaps $x_i \leftrightarrow x_{i+1}$ and $x_j \leftrightarrow x_{j+1}$ for $|i - j| \geq 2$. These constraints are equivalent to the requirement that $\{R_j\}_{j=1}^{n-1}$ generates a representation of the symmetric group $S_n$ (ref. 16). If this representation is not one-dimensional (1D), we say equation (3) defines a type of parastatistical particle, or paraparticles for short. Notice that the first relation in equation (4) is crucial for parastatistics to be consistently defined in any dimension; anyons generally do not

[1]Department of Physics and Astronomy, Rice University, Houston, TX, USA. [2]Smalley-Curl Institute, Rice University, Houston, TX, USA. [3]Max-Planck-Institut für Quantenoptik, Garching, Germany. ✉e-mail: zhiyuan.wang.physics@gmail.com

satisfy this relation and, consequently, they only form a representation of the braid group $B_n$ (ref. 16) instead of the symmetric group $S_n$ and are therefore limited to 2D (see Supplementary Information for a comparison between parastatistics and other known types of particle statistics).

Parastatistics, and their apparent absence in nature, has been discussed since the dawn of quantum mechanics[17]. The first concrete theory of parastatistics was proposed and investigated by Green in 1953 (ref. 6). This theory was subsequently studied in detail[11–15] and also more generally and rigorously[7–9,18] within the framework of algebraic quantum field theory[19,20]. These works did not rule out the existence of paraparticles in nature but led to the conclusion that, under certain assumptions, any theory of paraparticles (in particular, Green's theory) is physically indistinguishable from theories of ordinary fermions and bosons. This seemingly obviated the need to consider paraparticle theories, as they give exactly the same physical predictions as theories of ordinary particles.

In this paper, we show that non-trivial paraparticles inequivalent to either fermions or bosons exist in physical systems, in a way compatible with spatial locality and hermiticity. This poses no contradiction with earlier results, as the construction evades their restrictive assumptions. We demonstrate this by first introducing a second quantization formulation of parastatistics that is distinct from previous constructions (see Supplementary Information for a comparison), which includes exactly solvable theories of free paraparticles, and—in this formulation—paraparticles can show non-abelian permutation statistics (equation (3)) and generalized exclusion principles inequivalent to free fermions and bosons. Then we show that these paraparticles emerge as quasiparticle excitations in a family of exactly solvable quantum spin models, explicitly demonstrating how to avoid the aforementioned no-go theorems[7,8], allowing non-trivial consequences of parastatistics to be physically observed. Our second quantization formulation of paraparticles is valid in any spatial dimension and can be extended to incorporate special relativity, hinting at the potential existence of elementary paraparticles in nature.

## Basic formalism

We first present our second quantization formulation of parastatistics. This formulation only realizes a subfamily of the parastatistics defined by the first quantization approach presented above, but the pay-off is that it automatically guarantees the fundamental requirement of spatial locality, which is not ensured by the first quantization formulation. (See Section 3 in the Supplementary Information for the relation between the first and second quantization formulation of parastatistics in this paper). In this formulation, each type of parastatistics is labelled by a four-index tensor $R_{cd}^{ab}$ (in which $1 \le a, b, c, d \le m$, $m \in \mathbb{Z}$) satisfying

$$
\left[\begin{array}{c}{}^a R^b \\ {}_c R_d\end{array}\right] = \delta \left|\begin{array}{cc}a & b\end{array}\right| \delta \left|\begin{array}{cc}c & d\end{array}\right|, \quad \left[\begin{array}{ccc}a & b & c \\ R & & \\ & R & \\ d & e & f\end{array}\right] = \left[\begin{array}{ccc}a & b & c \\ & & R \\ R & & \\ d & e & f\end{array}\right], \tag{5}
$$

in which $R_{cd}^{ab} = {}_c^a\boxed{R}{}_d^b$ and, throughout this paper, we use tensor graphical notation, in which open indices are identified on both sides of the equation and contracted indices are summed over, and a line segment represents a Kronecker $\delta$ function. These two equations are reminiscent of equation (4) and we describe their precise relation in the Supplementary Information. The second equation in equation (5) is known in the literature as the constant Yang–Baxter equation (YBE)[21–23], whose solutions are called $R$ matrices. In Table 1, we present some basic examples of $R$ matrices and we can check by straightforward computation that they satisfy equation (5).

For a given $R$ matrix, we define the paraparticle creation and annihilation operators $\hat{\psi}_{i,a}^{\pm}$ through the commutation relations (CRs)

Nature | Vol 637 | 9 January 2025 | **315**

## Table 1 | Examples of $R$ matrices and their single-mode partition functions $z_R(x)$

| Ex. | 1 | 2 | 3 | 4 |
|---|---|---|---|---|
| $R_{cd}^{ab}$ | $-\delta_{ad}\delta_{bc}$ | $\delta_{ad}\delta_{bc}(-1)^{\delta_{ab}}$ | $-\delta_{ac}\delta_{bd}$ | $\lambda_{ab}\xi_{cd}-\delta_{ac}\delta_{bd}$ |
| $z_R(x)$ | $(1+x)^m$ | $(1+x)^m$ | $1+mx$ | $1+mx+x^2$ |

Here $z_R(x)$ is defined in equation (12), with $x=e^{-\beta\epsilon}$. The $\lambda$ and $\xi$ in Ex. 4 are $m\times m$ constant matrices satisfying $\lambda\xi\lambda^\mathsf{T}\xi^\mathsf{T}=\mathbb{1}_m$ and $\mathrm{Tr}[\lambda\xi^\mathsf{T}]=2$. (For example, we can take $\lambda=e^{-M}$, $\xi=-e^M$, in which $M$ is an $m\times m$ antisymmetric matrix, $M^\mathsf{T}=-M$, with complex entries satisfying $\mathrm{Tr}[e^{-2M}]=-2$. A small caveat here is that this example of $R$ matrix is not unitary for $m\ge3$ and the corresponding 1D spin model with emergent paraparticles defined later in the paper is not Hermitian but PT-symmetric; see the discussion in Section 4A of the Supplementary Information).

$$
\hat{\psi}_{i,a}^{-}\hat{\psi}_{j,b}^{+} = \sum_{cd} R_{bd}^{ac}\hat{\psi}_{j,c}^{+}\hat{\psi}_{i,d}^{-} + \delta_{ab}\delta_{ij},
$$
$$
\hat{\psi}_{i,a}^{+}\hat{\psi}_{j,b}^{+} = \sum_{cd} R_{ab}^{cd}\hat{\psi}_{j,c}^{+}\hat{\psi}_{i,d}^{+}, \tag{6}
$$
$$
\hat{\psi}_{i,a}^{-}\hat{\psi}_{j,b}^{-} = \sum_{cd} R_{dc}^{ba}\hat{\psi}_{j,c}^{-}\hat{\psi}_{i,d}^{-},
$$

in which $i$ and $j$ are mode indices (for example, position, momentum) and $a$, $b$, $c$ and $d$ are internal indices. Notice that $R_{cd}^{ab}=\pm\delta_{ad}\delta_{bc}$ gives back fermions (−) and bosons (+) with an internal degree of freedom. Although our construction works for any $R$ matrix satisfying equation (5), in this paper, we mainly focus on unitary $R$ matrices for simplicity, that is, $\sum_{a,b} R_{cd}^{ab}(R_{ef}^{ab})^* = \delta_{ce}\delta_{df}$, which is true for Exs. 1–3 in Table 1 and the $R$ matrix of the 2D solvable spin model (equations (29) and (31)). With a unitary $R$, we have $\hat{\psi}_{i,a}^{+} = (\hat{\psi}_{i,a}^{-})^\dagger$ (see Supplementary Information), which guarantees the hermiticity of physical observables, as we show later. (As we show in the Supplementary Information, even with a non-unitary $R$ matrix, such as Ex. 4 in Table 1, equation (6) is still consistently defined and most of the main results of this paper still apply).

A crucial structure in our construction is the Lie algebra of contracted bilinear operators defined as

$$
\hat{e}_{ij} \equiv \sum_{a=1}^{m} \hat{\psi}_{i,a}^{+}\hat{\psi}_{j,a}^{-}. \tag{7}
$$

We show that the space $\{\hat{e}_{ij}\}_{1\le i,j\le N}$ is closed under the commutator $[\hat{A},\hat{B}]=\hat{A}\hat{B}-\hat{B}\hat{A}$ and the corresponding Lie algebra is $\mathfrak{gl}_N$. First, using equation (6), we have

$$
[\hat{e}_{ij},\hat{\psi}_{k,b}^{+}] = \delta_{jk}\hat{\psi}_{i,b}^{+},
$$
$$
[\hat{e}_{ij},\hat{\psi}_{k,b}^{-}] = -\delta_{ik}\hat{\psi}_{j,b}^{-}, \tag{8}
$$

which leads to

$$
[\hat{e}_{ij},\hat{e}_{kl}] = \delta_{jk}\hat{e}_{il} - \delta_{il}\hat{e}_{kj}. \tag{9}
$$

(See Methods for detailed derivation). Equation (9) is the CR between the basis elements $\{\hat{e}_{ij}\}_{1\le i,j\le N}$ of the $\mathfrak{gl}_N$ Lie algebra, in which $\hat{e}_{ij}$ represents the matrix that has 1 in the $i$th row and $j$th column and 0 everywhere else. We will see that this Lie algebra structure enables straightforward construction of theories of paraparticles that obey locality, hermiticity and free-particle solvability.

In the usual case of fermions, physical observables are composed of even products of fermionic operators. This comes from the physical requirement of locality—local observables supported on disjoint regions (or space-like regions in relativistic quantum field theory) must commute. We define an analogue for parastatistics and show that they have analogous properties: for each local region of space $S$, we define a local observable on $S$ to be a Hermitian operator that is a sum of products of $\hat{e}_{ij}$, in which $i,j \in S$. For example, $\hat{O}_S = \hat{e}_{ij}\hat{e}_{ji}$ with $i,j \in S$ is a local

observable in $S$, as $\hat{e}_{ij}^{\dagger} = \hat{e}_{ji}$. Then, equation (9) immediately implies the aforementioned locality condition $[\hat{O}_{S_1}, \hat{O}_{S_2}] = 0$ for $S_1 \cap S_2 = \varnothing$. A locally interacting Hamiltonian $\hat{H}$ is defined to be a sum of local observables $\hat{H} = \sum_S h_S \hat{O}_S$, in which $h_S \in \mathbb{R}$ and the summation is over local regions $S$ whose diameters are smaller than some constant cut-off. This definition of local observables and Hamiltonians guarantees unitarity (time evolution $\hat{U} = e^{-iHt}$ generated by a Hamiltonian operator $\hat{H}$ is unitary) and microcausality (no signal can travel faster than a finite speed) in both relativistic quantum field theory and non-relativistic lattice quantum systems. For the former, the commutativity of local observables at space-like separations rules out faster-than-light travel and communication; for the latter, ref. 24 proved that, as long as all of the local Hamiltonian terms have uniformly bounded norms and their algebra has a local structure, then the Lieb–Robinson bound[25] holds, which gives an effective light cone of causality.

A particularly important family of physical observables is the particle number operators $\hat{n}_i \equiv \hat{e}_{ii}$. It follows from equation (9) that they mutually commute $[\hat{n}_i, \hat{n}_j] = 0$, so they have a complete set of common eigenstates. Meanwhile, equation (8) gives $[\hat{n}_i, \hat{\psi}_{j,b}^{\pm}] = \pm \delta_{ij} \hat{\psi}_{j,b}^{\pm}$, meaning that $\hat{\psi}_{j,b}^{+}$ ($\hat{\psi}_{j,b}^{-}$) increases (decreases) the eigenvalue of $\hat{n}_j$ by 1 and does not change the eigenvalue of $\hat{n}_i$ for $j \neq i$. This justifies the terminology creation and annihilation operators, because $\hat{\psi}_{j,b}^{+}$ ($\hat{\psi}_{j,b}^{-}$) creates (annihilates) a particle in the mode $j$. We also define the total particle number operator $\hat{n} = \sum_{i=1}^{N} \hat{n}_i$, so we have $[\hat{n}, \hat{\psi}_{i,b}^{\pm}] = \pm \hat{\psi}_{i,b}^{\pm}$. These CRs involving the number operators are the same as for fermions and bosons. However, we will see later that, owing to the generalized CRs between $\{\hat{\psi}_{i,b}^{+}\}$ in equation (6), the spectrum of $\{\hat{n}_i\}$ is different for paraparticles.

## Generalized exclusion statistics

Paraparticles defined by the CRs in equation (6) exhibit generalized exclusion statistics that is notably different from ordinary fermions and bosons. We demonstrate this phenomenon for the paraparticles defined by the $R$ matrix in Ex. 3 in Table 1 and present the general case in the Supplementary Information.

Analogous to the Fock space of fermions and bosons, there is a vacuum state $|0\rangle$ satisfying $\hat{\psi}_{i,a}^{-}|0\rangle = 0 \ \forall \ i, a$, so the vacuum contains no particles, $\hat{n}|0\rangle = 0$. The second line of equation (6) with $R_{cd}^{ab} = -\delta_{ac}\delta_{bd}$ (Ex. 3 in Table 1) reads

$$\hat{\psi}_{i,a}^{+}\hat{\psi}_{j,b}^{+} = -\hat{\psi}_{j,a}^{+}\hat{\psi}_{i,b}^{+} \ \forall \ i,j,a,b. \tag{10}$$

Taking $i = j$ in equation (10), we get $\hat{\psi}_{i,a}^{+}\hat{\psi}_{i,b}^{+} = 0$, which means that any mode $i$ cannot be occupied by two paraparticles even if they have different labels $a \neq b$, in contrast to fermions. Meanwhile, equation (10) does not imply any exclusion between paraparticles in different modes $i \neq j$, and the first line in equation (6) implies that the one-particle states $\hat{\psi}_{i,a}^{+}|0\rangle$ are orthonormal $\langle 0|\hat{\psi}_{j,b}^{-}\hat{\psi}_{i,a}^{+}|0\rangle = \delta_{ij}\delta_{ab}$. The whole state space is $(m+1)^N$-dimensional, spanned by orthonormal basis states of the form

$$|\Psi\rangle = \hat{\psi}_{i_1,a_1}^{+}\hat{\psi}_{i_2,a_2}^{+}...\hat{\psi}_{i_n,a_n}^{+}|0\rangle, \tag{11}$$

in which $0 \leq n \leq N$, $1 \leq i_1 < i_2 < ... < i_n \leq N$ and the action of $\hat{\psi}_{i,a}^{\pm}$ on these basis states is completely determined by the CRs in equation (6).

For a general $R$ matrix, a single mode $i$ can be occupied by several particles and the space of $n$-particle states $\hat{\psi}_{i,a_1}^{+}\hat{\psi}_{i,a_2}^{+}...\hat{\psi}_{i,a_n}^{+}|0\rangle$ is $d_n$-dimensional, in which $\{d_n\}_{n\geq 0}$ are non-negative integers that define the generalized exclusion statistics for the paraparticles associated with $R$, as shown in Fig. 1. In the above example, we have $d_0 = 1$, $d_1 = m$ and $d_n = 0 \ \forall \ n \geq 2$. This generalizes Fermi–Dirac statistics (in which $d_0 = d_1 = 1$ and $d_n = 0 \ \forall \ n \geq 2$) and Bose–Einstein statistics (in which $d_n = 1 \ \forall \ n \geq 0$). In the Supplementary Information, we show how to calculate $\{d_n\}_{n\geq 0}$ for a general $R$ matrix.

The numbers $\{d_n\}_{n\geq 0}$ allow us to compute the grand canonical partition function for a single mode at temperature $T$. Suppose that

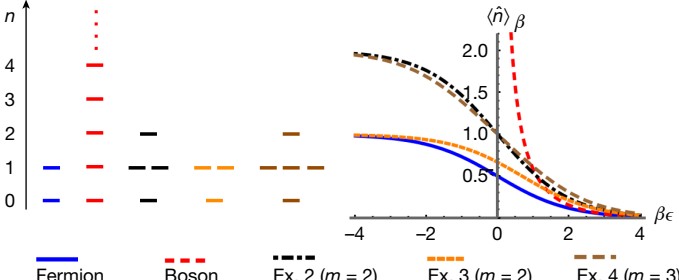

**Fig. 1 | Generalized exclusion statistics and thermodynamics of free paraparticles defined by the $R$ matrices and comparison with fermions and bosons.** The $R$ matrices are defined in Table 1. Left, the level degeneracy $\{d_n\}_{n\geq 0}$. Right, thermal expectation value of the single-mode occupation number $\langle \hat{n} \rangle_{\beta}$.

each particle in this mode carries energy $\epsilon$ (that is, the Hamiltonian is $\hat{H} = \epsilon\hat{n}$). Then

$$z_R(e^{-\beta\epsilon}) \equiv \text{Tr}[e^{-\beta\epsilon\hat{n}}] = \sum_{n=0}^{\infty} d_n e^{-n\beta\epsilon}, \tag{12}$$

in which $\beta = 1/(k_B T)$, $k_B$ is the Boltzmann constant and we have absorbed the chemical potential $\mu$ into $\epsilon$. The single-mode partition functions $z_R(e^{-\beta\epsilon})$ for the $R$ matrices in Exs. 1–4 are given in Table 1. Multimode partition functions factorize into products of single-mode partition functions exactly as for fermions and bosons.

The single-mode partition function $z_R(x)$ (in which $x = e^{-\beta\epsilon}$) provides a straightforward demonstration of the non-triviality (that is, distinct from fermions and bosons) of the parastatistics for some $R$ matrices. If the paraparticle system defined by $R$ can be transformed into a system of $p$ flavours of free fermions and $q$ flavours of free bosons, then $z_R(x) = (1+x)^p(1-x)^{-q}$. Therefore the $R$ matrix given in Ex. 3 must define a non-trivial type of parastatistics for $m \geq 2$, as $z_R(x) = 1 + mx$ is not equal to $(1+x)^p(1-x)^{-q}$ for any integers $p, q$. Example 4 is similarly non-trivial for $m \geq 3$ (A caveat, however, is that $z_R(x)$ only gives a sufficient condition for non-triviality and having a trivial $z_R(x)$ does not imply that the corresponding paraparticle theory is completely equivalent to fermions or bosons. For example, the emergent paraparticles in our 2D solvable spin model are defined by the $R$ matrix given in equations (29) and (31), whose partition function $z_R(x) = (1+x)^4$ is the same as free fermions with an SU(4) symmetry, but the exchange statistics of these emergent paraparticles are still physically distinct from fermions and boson, as we show in Methods).

## Particle exchange statistics

In addition to generalized exclusion statistics, paraparticles defined by equation (6) also show exotic exchange statistics defined by the $R$ matrix that results from physically exchanging paraparticles. Consider a state with two paraparticles at different positions $i \neq j$: $|0; ia, jb\rangle \equiv \hat{\psi}_{i,a}^{+}\hat{\psi}_{j,b}^{+}|0\rangle$. Let $\hat{E}_{ij}$ be a unitary operator that exchanges the positions of the paraparticles at $i$ and $j$:

$$\hat{E}_{ij}\hat{\psi}_{i,a}^{+}\hat{E}_{ij}^{\dagger} = \hat{\psi}_{j,a}^{+}, \quad \hat{E}_{ij}\hat{\psi}_{j,a}^{+}\hat{E}_{ij}^{\dagger} = \hat{\psi}_{i,a}^{+} \quad \forall \ a. \tag{13}$$

Note that such an operator can always be constructed from a product of local unitaries of the form $e^{i\frac{\pi}{2}(\hat{e}_{kl}+\hat{e}_{lk})}$, which exchanges $\hat{\psi}_{k,a}^{+} \leftrightarrow \hat{\psi}_{l,a}^{+}$. The exchange operator $\hat{E}_{ij}$ acts on the two particle states $|0; ia, jb\rangle$ as

$$\hat{E}_{ij}|0; ia, jb\rangle = (\hat{E}_{ij}\hat{\psi}_{i,a}^{+}\hat{E}_{ij}^{\dagger})(\hat{E}_{ij}\hat{\psi}_{j,b}^{+}\hat{E}_{ij}^{\dagger})\hat{E}_{ij}|0\rangle$$
$$= \hat{\psi}_{j,a}^{+}\hat{\psi}_{i,b}^{+}|0\rangle \tag{14}$$
$$= \sum_{a',b'} R_{ab}^{b'a'}|0; ib', ja'\rangle,$$

in which in the second line we applied equation (13) and the invariance of $|0\rangle$ under $\hat{E}_{ij}$ and in the third line we used the fundamental CR (equation (6)) between $\hat{\psi}_{j,a}^{+}$ and $\hat{\psi}_{i,a}^{+}$. Equation (14) defines the physical meaning of the $R$ matrix as the unitary rotation of the two-particle state space that results from physically exchanging paraparticles and, in the solvable spin models with emergent paraparticles we present later, the effect of such a unitary rotation can be directly explored using local operations and measurements, which shows a substantial difference from ordinary fermions and bosons, as we show in Methods. The above derivation is valid in any spatial dimension and can be directly generalized to states with many paraparticles.

## Exact solution of free paraparticles

In our second quantization framework, the general bilinear Hamiltonian describing free paraparticles,

$$\hat{H} = \sum_{1 \leq i,j \leq N} h_{ij} \hat{e}_{ij} = \sum_{\substack{1 \leq i,j \leq N \\ 1 \leq a \leq m}} h_{ij} \hat{\psi}_{i,a}^{+} \hat{\psi}_{j,a}^{-}, \tag{15}$$

can be solved analogously to bosons and fermions. We outline this here but further details can be found in Methods. We require $h_{ij}^{*} = h_{ji}$ so that $\hat{H}^{\dagger} = \hat{H}$. Using a canonical transformation of $\{\hat{\psi}_{i,a}^{\pm}\}$, the Hamiltonian becomes $\hat{H} = \sum_{k=1}^{N} \epsilon_{k} \tilde{n}_{k}$, in which $\{\epsilon_{k}\}_{k=1}^{N}$ are the eigenvalues of the coefficient matrix $h_{ij}$ and $\{\tilde{n}_{k}\}_{k=1}^{N}$ are mutually commuting occupation number operators for each mode $k$. The partition function of the whole system, $\mathrm{Tr}[e^{-\beta \hat{H}}]$, factorizes as a product of single-mode partition functions in equation (12), from which we obtain the average occupation number of mode $k$

$$\langle \tilde{n}_{k} \rangle_{\beta} \equiv \frac{\mathrm{Tr}[\tilde{n}_{k} e^{-\beta \hat{H}}]}{\mathrm{Tr}[e^{-\beta \hat{H}}]} = \frac{z_{R}'(e^{-\beta \epsilon_{k}}) e^{-\beta \epsilon_{k}}}{z_{R}(e^{-\beta \epsilon_{k}})}. \tag{16}$$

Figure 1 plots $\langle \tilde{n}_{k} \rangle_{\beta}$ as a function of $\beta \epsilon_{k}$ for the $R$ matrices in Exs. 3 and 4 (Table 1) with $m = 5$, showing the distinct finite-temperature thermodynamics of paraparticles compared with ordinary fermions and bosons, characterizing a new type of ideal gas.

## Emergent paraparticles in condensed matter systems

Finally, we discuss the potential impacts of paraparticles, including routes to observe them in nature, starting with the promising setting for paraparticles as quasiparticle excitations in condensed matter systems. Substantial insight in this direction and a proof of principle that such excitations can occur in physical systems are provided by a family of exactly solvable quantum spin systems, in which free paraparticles emerge as quasiparticle excitations. Here we present the 1D case for simplicity and we also discuss a 2D model whose details are presented in Methods and the Supplementary Information. For each $R$ matrix, we define a Hamiltonian

$$\hat{H} = \sum_{i,a} J_{i} (\hat{x}_{i,a}^{+} \hat{y}_{i+1,a}^{-} + \hat{x}_{i,a}^{-} \hat{y}_{i+1,a}^{+}) - \sum_{i,a} \mu_{i} \hat{y}_{i,a}^{+} \hat{y}_{i,a}^{-}, \tag{17}$$

in which $\{\hat{x}_{i,a}^{\pm}, \hat{y}_{i,a}^{\pm}\}_{a=1}^{m}$ are local spin operators (that is, operators on different sites commute) acting on the $i$th site, whose definition depends on the $R$ matrix. The index $i$ runs from 1 to $N$, with $N$ being the system size, and we use open boundary condition $J_{N} = 0$. The model has a total conserved charge $\hat{n} = \sum_{i,a} \hat{y}_{i,a}^{+} \hat{y}_{i,a'}^{-}$, which will be mapped to the paraparticle number operator, and $\hat{x}_{i,a}^{+}, \hat{y}_{i,a}^{+} (\hat{x}_{i,a}^{-}, \hat{y}_{i,a}^{-})$ increase (decrease) $\hat{n}$ by 1. For example, with the $R$ matrix in Ex. 3, the local Hilbert space $\mathfrak{V}$ is $m + 1$-dimensional, with basis states $|0\rangle, \{|1, b\rangle\}_{b=1}^{m}$, the $\hat{y}_{a}^{\pm}$ are defined as (omitting the site label) $\hat{y}_{a}^{+}|0\rangle = |1, a\rangle$, $\hat{y}_{a}^{-}|1, b\rangle = \delta_{ab}|0\rangle$, $\hat{y}_{a}^{-}|0\rangle = \hat{y}_{a}^{+}|1, b\rangle = 0$ and $\hat{x}_{a}^{\pm} = \hat{y}_{a}^{\pm}$. This is a simple, nearest-neighbour spin model that is

realized in three-level Rydberg atom or molecule systems[26,27]. For the definition of $\hat{x}_{a}^{\pm}$ and $\hat{y}_{a}^{\pm}$ in general, see the Supplementary Information.

This model can be solved using a substantial generalization of the Jordan–Wigner transformation (JWT)[28] that we introduce here, in which the products of operators ('strings') are replaced with MPOs[29]. Specifically, we introduce operators

$$\tag{18}$$

in which $\hat{y}_{j,a}^{\pm} \equiv a \langle\!\!\!\triangleleft|_{j}$ and $\hat{T}_{j,ab}^{\pm} \equiv a \langle\!\!\!\!\Diamond|b = \mp [\hat{y}_{j,a}^{\pm}, \hat{x}_{j,b}^{\mp}]$ are local spin operators acting on site $j$. Both $\hat{\psi}_{i,a}^{\pm}$ act non-trivially on sites $1, 2, \ldots, i$ and act as identity on the rest of the chain. For example, with the $R$ matrix in Ex. 3, $\hat{T}_{ab}^{+}$ act as $\hat{T}_{ab}^{+}|0\rangle = \delta_{ab}|0\rangle$, $\hat{T}_{ab}^{-}|1, c\rangle = -\delta_{ac}|1, b\rangle$ and $\hat{T}_{ab}^{+}|1, c\rangle = -\delta_{bc}|1, a\rangle$. In the special case $m = 1, R = -1$, $\hat{H}$ in equation (17) is the Hamiltonian for the spin-1/2 XY model, the operators $\hat{\psi}_{i,a}^{\pm}$ are fermion creation and annihilation operators and the MPO JWT simplifies to the ordinary JWT.

The $\hat{\psi}_{i,a}^{\pm}$ constructed in equation (18) satisfy the parastatistical CRs in equation (6), as we prove in the Supplementary Information using tensor network manipulations. Moreover, the Hamiltonian in equation (17) can be rewritten in terms of $\{\hat{\psi}_{i,a}^{\pm}\}$ as

$$\hat{H} = \sum_{i,a} J_{i} (\hat{\psi}_{i,a}^{+} \hat{\psi}_{i+1,a}^{-} + \hat{\psi}_{i+1,a}^{+} \hat{\psi}_{i,a}^{-}) - \sum_{i} \mu_{i} \hat{n}_{i}, \tag{19}$$

therefore $\hat{\psi}_{i,a_1}^{\pm}$ create/annihilate free emergent paraparticles. Using a canonical transformation of $\{\hat{\psi}_{i,a}^{\pm}\}$, the free paraparticle Hamiltonian in equation (19) can be diagonalized into the form $\hat{H} = \sum_{k=1}^{N} \epsilon_{k} \tilde{n}_{k}$ and all energy eigenvalues can be exactly obtained for arbitrary coupling constants $\{J_{i}\}$ (even with disorder).

Exactly solvable quantum spin models with free emergent paraparticles can also be found in 2D. In Methods, we present the key features of these models through a specific example with $m = 4$. These 2D models realize a special family of paraparticles that, despite having trivial exclusion statistics (that is, the same partition function as $m$ flavours of fermions), have non-trivial exchange statistics that is physically (observably) distinct from fermions and bosons. Similar to the 1D case, these models are mapped to free paraparticle Hamiltonians of the form in equation (15), using a MPO JWT defined in equation (34) that generalizes equation (18). In 2D, $\hat{\psi}_{i,a_1}^{\pm}$ are still MPO string operators, with the further notable property that their actions on the low-energy sector (for example, the ground states) are independent of the paths on which they are defined, which is reminiscent of the path independence property of the string (ribbon) operators that create anyons in Kitaev's quantum double model[30].

In summary, these results imply a new type of quasiparticle statistics, which can be searched for in condensed matter systems, and a starting point is the exactly solvable quantum spin model defined in equation (17) and its 2D generalizations, defined in equation (32). Systems with such excitations may show a wealth of new phenomena and the exactly solvable models constructed above provide an efficient way to study them. Depending on the resulting free paraparticle systems to which the spin models are mapped, new phases of matter and phase transitions can be discovered. For example, in 2D, if the free paraparticle system has a non-trivial topological band structure (having a non-zero Chern number), then the spin model can be in a new chiral topological phase that is hard to study with previous techniques (The chiral topological phases of our 2D models are expected to lie beyond those found in previous solvable models[31–33], as explained in Methods. The study of chiral topological order using tensor network techniques is also known to be hard[34].). If the free paraparticle system has a gapless spectrum, the

spin model can realize a phase transition point or a gapless topological phase[35–37], which are interesting and difficult areas of research, even in 1D systems. Furthermore, allowing the tunnelling constants $\{J_i\}$ to be spatially disordered may lead to new localized phases.

## Speculations about elementary paraparticles

In addition to the possibility of emergent parastatistical excitations in interacting quantum matter, a natural, albeit highly speculative, question is to ask whether paraparticles may exist as elementary particles in nature. We have seen that our second quantized theory of paraparticles satisfies the fundamental requirements of locality and hermiticity and is consistently defined in all dimensions. It is also straightforward to incorporate relativity to obtain a fully consistent relativistic quantum field theory of elementary paraparticles, in which the canonical quantization of field operators is defined by the CRs in equation (6). Most fundamental field-theoretical concepts and tools[38] generalize straightforwardly to parastatistics.

To consider paraparticles as elementary particles, it is important to consider their superselection rules. We discuss this issue in Section 6 of the Supplementary Information, in which we explain how superselection rules fundamentally constrain the observability of parastatistics, which is reminiscent of the previous no-go theorems[7,8]. We then discuss how our proposed realization of emergent paraparticles in condensed matter systems breaks these superselection rules, which motivates routes to construct theories of elementary paraparticles observably distinct from fermions and bosons, evading the no-go theorems[7,8].

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

## Methods

### Derivations

Derivations for equations (8) and (9) were as follows. The commutator between $\hat{e}_{ij}$ and $\hat{\psi}^{\pm}_{k,b}$ is

$$[\hat{e}_{ij}, \hat{\psi}^{+}_{k,b}] = \sum_a (\hat{\psi}^{+}_{i,a}\hat{\psi}^{-}_{j,a}\hat{\psi}^{+}_{k,b} - \hat{\psi}^{+}_{k,b}\hat{\psi}^{+}_{i,a}\hat{\psi}^{-}_{j,a})$$

$$= \sum_a \hat{\psi}^{+}_{i,a}\left(\sum_{c,d} R^{ac}_{bd}\hat{\psi}^{+}_{k,c}\hat{\psi}^{-}_{j,d} + \delta_{jk}\delta_{ab}\right)$$

$$\quad - \sum_a \hat{\psi}^{+}_{k,b}\hat{\psi}^{+}_{i,a}\hat{\psi}^{-}_{j,a}$$

$$= \sum_{a,c,d} (R^{ac}_{bd}\hat{\psi}^{+}_{i,a}\hat{\psi}^{+}_{k,c})\hat{\psi}^{-}_{j,d} + \delta_{jk}\hat{\psi}^{+}_{i,b}$$

$$\quad - \sum_a \hat{\psi}^{+}_{k,b}\hat{\psi}^{+}_{i,a}\hat{\psi}^{-}_{j,a}$$

$$= \delta_{jk}\hat{\psi}^{+}_{i,b}, \tag{20}$$

in which in the second (third) line we used the first (second) line of equation (6). Similarly, we have

$$[\hat{e}_{ij}, \hat{\psi}^{-}_{k,b}] = -\delta_{ik}\hat{\psi}^{-}_{j,b}. \tag{21}$$

Now we can compute the commutator

$$[\hat{e}_{ij}, \hat{e}_{kl}] = \sum_b [\hat{e}_{ij}, \hat{\psi}^{+}_{k,b}]\hat{\psi}^{-}_{l,b} + \sum_b \hat{\psi}^{+}_{k,b}[\hat{e}_{ij}, \hat{\psi}^{-}_{l,b}]$$

$$= \sum_b \delta_{jk}\hat{\psi}^{+}_{i,b}\hat{\psi}^{-}_{l,b} - \sum_b \hat{\psi}^{+}_{k,b}\delta_{il}\hat{\psi}^{-}_{j,b}$$

$$= \delta_{jk}\hat{e}_{il} - \delta_{il}\hat{e}_{kj}, \tag{22}$$

in which in the second line we used equations (20) and (21).

### Exact solution of free paraparticles

Here we present details for solving the general bilinear Hamiltonian in equation (15). Analogous to usual free bosons and fermions, we consider $U(N)$ transformations of $\{\hat{\psi}^{\pm}_{i,a}\}$:

$$\hat{\psi}^{-}_{i,a} = \sum_{k=1}^{N} U^*_{ki}\widetilde{\psi}^{-}_{k,a},$$

$$\hat{\psi}^{+}_{i,a} = \sum_{k=1}^{N} U_{ki}\widetilde{\psi}^{+}_{k,a}, \tag{23}$$

in which $U_{ki}$ is an $N \times N$ unitary matrix and we use operators with a tilde $\widetilde{\psi}^{\pm}_{k,a}$ to denote eigenmode operators. Inserting equation (23) into equation (6), we see that the operators $\{\widetilde{\psi}^{\pm}_{k,a}\}$ satisfy exactly the same CRs as $\{\hat{\psi}^{\pm}_{i,a}\}$. Notice that most of our discussions on the second quantization formulation and the state space only assume the CRs in equation (6), so the results obtained for $\{\hat{\psi}^{\pm}_{i,a}\}$ (in particular, the Lie algebra of bilinear operators and the structure of the state space) must also apply to $\{\widetilde{\psi}^{\pm}_{k,a}\}$.

Inserting equation (23) into equation (15), we obtain

$$\hat{H} = \sum_{\substack{1 \le k,p \le N \\ 1 \le a \le m}} h'_{kp}\widetilde{\psi}^{+}_{k,a}\widetilde{\psi}^{-}_{p,a} \equiv \sum_{1 \le k,p \le N} h'_{kp}\tilde{e}_{kp}, \tag{24}$$

in which $h'_{kp} = \sum_{1 \le i,j \le N} U_{ki}h_{ij}U^*_{pj} = [UhU^\dagger]_{kp}$. We can therefore choose the unitary matrix $U$ such that $h'_{kp} = \epsilon_k\delta_{kp}$, in which $\{\epsilon_k\}_{k=1}^N$ are eigenvalues of $h_{ij}$. With this choice, the Hamiltonian becomes diagonal $\hat{H} = \sum_{k=1}^N \epsilon_k\tilde{n}_k$ and its eigenstates can be taken as the common eigenstates

$\left|\begin{smallmatrix} \alpha_1, & \alpha_2, & \dots, & \alpha_N \\ \tilde{n}_1, & \tilde{n}_2, & \dots, & \tilde{n}_N \end{smallmatrix}\right\rangle$ (defined in equation (S4) of the Supplementary Information) of the mutually commuting operators $\{\tilde{n}_k\}_{k=1}^N$, with energy eigenvalues $E = \sum_{k=1}^N \epsilon_k\tilde{n}_k$, in which $\{\tilde{n}_k\}_{k=1}^N$ are independent non-negative integers and $1 \le \alpha_k \le d_{\tilde{n}_k}$ encodes the single-particle exclusion statistics.

We now calculate physical observables at temperature $T$. The partition function is a product of single-mode partition functions in equation (12)

$$Z(\beta) \equiv \mathrm{Tr}[e^{-\beta\hat{H}}] = \prod_k z_R(e^{-\beta\epsilon_k}), \tag{25}$$

so the free energy is

$$F(\beta) = -\frac{1}{\beta}\ln Z(\beta) = -\frac{1}{\beta}\sum_k \ln z_R(e^{-\beta\epsilon_k}). \tag{26}$$

The partition function allows us to compute the thermal average of observables $\tilde{n}^l_k$ and $\tilde{e}_{kp}$

$$\langle \tilde{n}^l_k \rangle_\beta = \frac{\mathrm{Tr}[\tilde{n}^l_k e^{-\beta\hat{H}}]}{\mathrm{Tr}[e^{-\beta\hat{H}}]} = \frac{(x\partial_x)^l z_R(x)}{z_R(x)}\bigg|_{x=e^{-\beta\epsilon_k}},$$

$$\langle \tilde{e}_{kp} \rangle_\beta = \delta_{kp}\langle \tilde{n}_k \rangle_\beta. \tag{27}$$

The thermal average for physical operators $\hat{e}_{ij}$ are obtained by transforming creation and annihilation operators to the eigenmode basis using equation (23) and using the result for $\langle \tilde{e}_{kp} \rangle_\beta$ given in equation (27), which yields

$$\langle \hat{e}_{ij} \rangle_\beta = \sum_k U_{ki}U^*_{kj}\langle \tilde{n}_k \rangle_\beta. \tag{28}$$

The thermal average for other physical observables, including correlation functions in and out of equilibrium, can all be calculated exactly in a similar way.

### R matrix for the 2D solvable spin model

In the following, we present a unitary $R$ matrix with trivial exclusion statistics but non-trivial exchange statistics, on which the 2D solvable spin model is based. We define the $R$ matrix with $m = 4$ in the following way. Let $S = \{1, 2, 3, 4\}$ and $r: S \times S \to S \times S$ be an injective map defined as

$$r(a,b) = \begin{pmatrix} 43 & 12 & 24 & 31 \\ 21 & 34 & 42 & 13 \\ 14 & 41 & 33 & 22 \\ 32 & 23 & 11 & 44 \end{pmatrix}_{ab}, \tag{29}$$

in which we use $ab$ as a shorthand for $(a, b)$. For example, $r(1, 1) = (4, 3)$ and $r(3, 2) = (4, 1)$. The map $r$ in equation (29) satisfies the set-theoretical YBE[23]

$$r^2 = \mathrm{id}_{S\times S}, \quad r_{12}r_{23}r_{12} = r_{23}r_{12}r_{23}, \tag{30}$$

in which in the second equation both sides are injective maps from the set $S \times S \times S$ to itself, $r_{12} = r \times \mathrm{id}_S$ and $r_{23} = \mathrm{id}_S \times r$. Now we define the $R$ matrix as

$$R|a,b\rangle = -|b',a'\rangle \quad \forall a,b \in S, \tag{31}$$

in which $(b', a') = r(a, b)$. It then follows from equation (30) that $R$ satisfies the YBE (equation (5)). The single-mode partition function $z_R(x)$ of this $R$ matrix is $z_R(x) = (1 + x)^4$ (see Supplementary Information), meaning that the exclusion statistics of this type of paraparticles is the same as four decoupled flavours of ordinary fermions. Despite having trivial exclusion statistics, the permutation statistics defined

by this $R$ matrix is notably distinct from fermions, as is manifest in the paraparticle exchange process in the 2D solvable spin model that we demonstrate later.

## Solvable 2D spin models with emergent free paraparticles

In the following, we present a solvable 2D quantum spin model with emergent free paraparticles, based on the set-theoretical $R$ matrix in equation (29). Here we only sketch the key definitions and the main results; the technical details are found in the Supplementary Information. The model is defined on a square lattice with two types of lattice site and open boundary condition, as shown in Extended Data Fig. 1. The Hamiltonian consists of two parts, $\hat{H} = \hat{H}_1 + \hat{H}_2$,

$$
\begin{aligned}
\hat{H}_1 &= \sum_v \hat{A}_v + \sum_p \hat{B}_p, \\
\hat{H}_2 &= -\sum_{\langle ij \rangle} \hat{h}_{ij} - \sum_l \mu \hat{y}_{l,a}^+ \hat{y}_{l,a}^-,
\end{aligned}
\tag{32}
$$

in which $v$ and $p$ denote the shaded and white plaquettes, respectively, $l$ runs over all black dots and $\langle ij \rangle$ runs over all neighbouring pairs of black dots (each pair appears only once). Here $\hat{h}_{ij}$ is a three-body interaction between the vertices of the triangle containing the directed edge $\langle ij \rangle$, defined as

$$
k \circ\!\!\!\triangleleft\!\!\!\overset{j}{\underset{i}{w}} = J_{ij} \;\triangleright\!\!\!\overset{w^+}{\underset{i\;\;k}{}}\!\!\!\triangleleft^+_j + \text{h.c.} \equiv \hat{h}_{ij},
\tag{33}
$$

in which $\hat{y}_{j,a}^{\pm} \equiv {}^a\!\triangleleft_j$ and $\hat{x}_{j,a}^{\pm} \equiv \triangleright^a$ are the same spin operators that appeared in the 1D model, $w$ is one of $u_L$, $u_R$, $v_L$ or $v_R$, depending on the type of triangle in the lattice and $\hat{w}_{ab}^{\pm} = {}^a\!\langle w^{\pm}\rangle\, b$ is an operator acting on an auxiliary site (open circles in Extended Data Fig. 1), for $a, b = 1, ..., 4$. The definition of the tensors $u_L^{\pm}$, $u_R^{\pm}$, $v_L^{\pm}$ and $v_R^{\pm}$ are given in the Supplementary Information. The operators $\hat{A}_v$ and $\hat{B}_p$ in equation (32) are eight-body interaction terms defined as

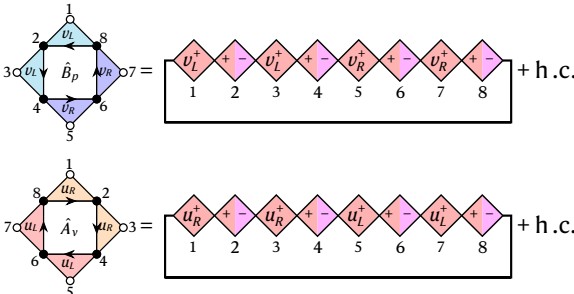

(If a loop term lies on the boundary, then one or more of its white circles will be absent. In this case, the tensors $u_L^{\pm}$, $u_R^{\pm}$, $v_L^{\pm}$ and $v_R^{\pm}$ on the absent site is replaced by a $\delta$ tensor, that is, $\hat{w}_{ab}^{\pm} = \delta_{ab}$, for $w = u_L$, $u_R$, $v_L$ and $v_R$).

The loop terms $\hat{A}_v$ and $\hat{B}_p$ are constructed such that they mutually commute and commute with each individual three-body term in $\hat{H}_2$; therefore, they are conserved quantities and eigenstates of $\hat{H}$ can be labelled by their common eigenvalues. In this paper, we are mainly interested in the subspace of states in which all $\hat{A}_v$ and $\hat{B}_p$ have minimal eigenvalues (that is, the space of ground states of $\hat{H}_1$), henceforth referred to as the zero-vortex sector $\Phi_0$. The Hilbert space dimension of this sector is $16^N$, in which $N$ is the total number of black dots in the lattice.

To solve the spectrum in the zero-vortex sector, we define paraparticle creation and annihilation operators by means of a generalized MPO JWT, which generalizes the 1D case given in equation (18). Each paraparticle operator $\hat{\psi}_{i,a}^{\pm}$ is defined on a string $\Gamma$ connecting the lattice

origin to the site $i$, and $\hat{\psi}_{i,a}^{\pm}$ is a MPO acting consecutively on all of the black dots on $\Gamma$ (including the start and end points) and all of the open circles adjacent to $\Gamma$; see Extended Data Fig. 1 for an example. The MPO representation of $\hat{\psi}_{i,a}^{\pm}$ is similar to the 1D case given in equation (18), but now with the tensors $w^{\pm}$ inserted between neighbouring $T^{\pm}$, in which $w$ is one of $u_L$, $v_L$, $u_R$ or $v_R$, depending on the type of triangle between the neighbouring black dots. For example, for the string $\Gamma$ in Extended Data Fig. 1 that starts at point 0 (lattice origin) and ends at point $i$, $\hat{\psi}_{i,a}^{\pm}$ acts on all of the purple dots and is defined as

$$
\begin{aligned}
\hat{\psi}_{i,a}^+ &= a \underset{0}{\triangleleft} \overset{-}{\underset{1}{+}} \underset{2}{v_R} \overset{+}{\underset{4}{-}} \overset{-}{\underset{5}{+}} \underset{}{u_R} \overset{+}{\underset{6}{-}} \overset{-}{\underset{7}{+}} \underset{}{v_L} \overset{+}{\underset{i}{\triangleright}}, \\
\hat{\psi}_{i,a}^- &= a \underset{0}{\triangleleft} \overset{+}{\underset{1}{-}} \underset{2}{v_R} \overset{-}{\underset{4}{+}} \overset{+}{\underset{5}{-}} \underset{}{u_R} \overset{-}{\underset{6}{+}} \overset{+}{\underset{7}{-}} \underset{}{v_L} \overset{-}{\underset{i}{\triangleright}}.
\end{aligned}
\tag{34}
$$

Paraparticle operators $\{\hat{\psi}_{i,a}^{\pm}\}$ constructed this way have several important properties. First, they commute with all individual terms in $\hat{H}_1$, therefore, their actions leave the zero-vortex sector $\Phi_0$ invariant. Second, as shown in the Supplementary Information, although each paraparticle operator $\hat{\psi}_{i,a}^{\pm}$ is defined on a specific path, their actions in the zero-vortex sector $\Phi_0$ do not depend on the choice of the path, only on the end points. This is because of the special topological property of the zero-vortex sector and is reminiscent of the path independence of the action of the string operators on the toric code ground states[30]. Finally, in the zero-vortex sector, the operators $\{\hat{\psi}_{i,a}^{\pm}\}$ satisfy the parastatistical CRs in equation (6), justifying their name 'paraparticle operators'. These properties lead us to Theorem 1 (see also Supplementary Information).

**Theorem 1.** In the zero-vortex sector, $\hat{H}_2$ is mapped to the free paraparticle Hamiltonian

$$
\hat{H}_2 = -\sum_{\langle ij \rangle, 1 \le a \le m} (J_{ij} \hat{\psi}_{j,a}^+ \hat{\psi}_{i,a}^- + \text{h.c.}) - \sum_l \mu_l \hat{n}_l.
\tag{35}
$$

We expect that our 2D solvable spin models exhibit new chiral and gapless topological phases that are not exhibited by previous solvable models. So far, the only family of solvable models for chiral topological order in 2D is Kitaev's honeycomb model[31] and its generalizations[32,33], whose gapped phases are classified by the 16-fold way[31], depending on the Chern number ($\nu \bmod 16$) of the free fermion band. We expect that the gapped phases of our model are similarly classified by the Chern number of the free paraparticle band. When $\nu = 0$, both Kitaev's honeycomb model and our models are in non-chiral quantum double phases, but the former only hosts $\mathbb{Z}_2$ abelian anyons, whereas the latter host non-abelian anyons already at $\nu = 0$. We expect that our models host different chiral topological phases also at non-zero $\nu$ and different gapless topological phases when the free paraparticles have a gapless spectrum.

## Particle exchange statistics in the 2D solvable model

We now illustrate the exchange statistics of the emergent paraparticles in the 2D solvable spin model, which reveals a notable physical difference between the emergent paraparticles and ordinary fermions and bosons.

Consider the paraparticle exchange process described in Extended Data Fig. 2. For simplicity, we consider the case when $-\mu_l$ is large, so that the ground state $|G\rangle$ of the 2D system has no paraparticles, that is, $\hat{n}_l |G\rangle = 0\ \forall\ l$. At $t = 0$, we can apply local unitary operators on the ground state $|G\rangle$ to create a paraparticle at sites $i$ and $j$, respectively, and obtain the state $|G; ia, jb\rangle \equiv \hat{\psi}_{i,a}^+ \hat{\psi}_{j,b}^+ |G\rangle$ (see Supplementary Information). Then we evolve the state $|G; ia, jb\rangle$ with $\hat{E}_{ij}$, which moves the paraparticles along the coloured paths shown in Extended Data Fig. 2 ($\hat{E}_{ij}$ can be constructed from a product of local unitaries of the form $e^{i\frac{\pi}{2}(\hat{e}_{kl} + \hat{e}_{lk})}$, in which $\hat{e}_{kl}$ is mapped to a local three-body interaction

in the 2D model). The result of this unitary exchange process is given by equation (14), in which $|0\rangle$ is understood as the ground state $|G\rangle$. With the set-theoretical $R$ matrix in equations (29) and (31), the final state is $-|G; ib', ja'\rangle$, in which $(b', a') = r(a, b)$, and the labels $a'$ and $b'$ can be locally measured at the two corners (See Supplementary Information). For example, if we start with $a = b = 1$, we end up measuring $b' = 4$ and $a' = 3$. That is, the auxiliary space of the paraparticles undergoes a non-trivial unitary rotation even though the two particles stay arbitrarily far apart from each other throughout the whole process. This is in contrast with fermions and bosons, in which case we would measure $a' = a$ and $b' = b$, that is, the indices are simply carried with the particles without any change.

In principle, the exchange process described above can also be done in the 1D spin model in equation (17). In this case, the paraparticles can also be created and measured at the two ends of the open chain, equation (14) still holds and the measurement result is the same. The main difference from the 2D case is that, in 1D, the two paraparticles inevitably collide during the exchange and the exchange statistics results from the interaction between the two paraparticles, which is sensitive to the microscopic details of the exchange operator $\hat{E}_{ij}$ and is not robust against local perturbations. By contrast, in 2D, the paraparticles can stay far away from each other throughout the exchange and their exchange statistics has a topological nature independent of the detailed shape of the space-time trajectory of the particles and is robust against all local perturbations when the particles are far away from the boundaries and from each other.

## Code availability

Mathematica codes for verifying some technical details of this paper are available at https://github.com/lagrenge94/Mathematica-codes-for-parastatistics. All of the algebraic data used in this paper are provided in the same package.

**Acknowledgements** We thank A. Kitaev, J. I. Cirac, K. Slagle, A. Long, M. Amin, A. Hahn and P. Fendley for discussions. We acknowledge support from the Robert A. Welch Foundation (C-1872), the National Science Foundation (PHY-1848304), the Office of Naval Research (N00014-20-1-2695) and the W. M. Keck Foundation (grant no. 995764). The contribution of K.R.A.H. benefited from discussions at the Aspen Center for Physics, supported by the National Science Foundation under grant no. PHY1066293, and the KITP, which was supported in part by the National Science Foundation under grant no. NSF PHY1748958. Z.W. is supported by the Munich Quantum Valley (MQV), which is supported by the Bavarian state government with funds from the Hightech Agenda Bayern Plus.

**Author contributions** Z.W. proposed the mathematical framework in this paper under the supervision of K.R.A.H. and both authors contributed extensively in interpreting its physics and in writing the paper.

**Funding** Open access funding provided by Max Planck Society.

**Competing interests** The authors declare no competing interests.

**Additional information**
**Correspondence and requests for materials** should be addressed to Zhiyuan Wang.

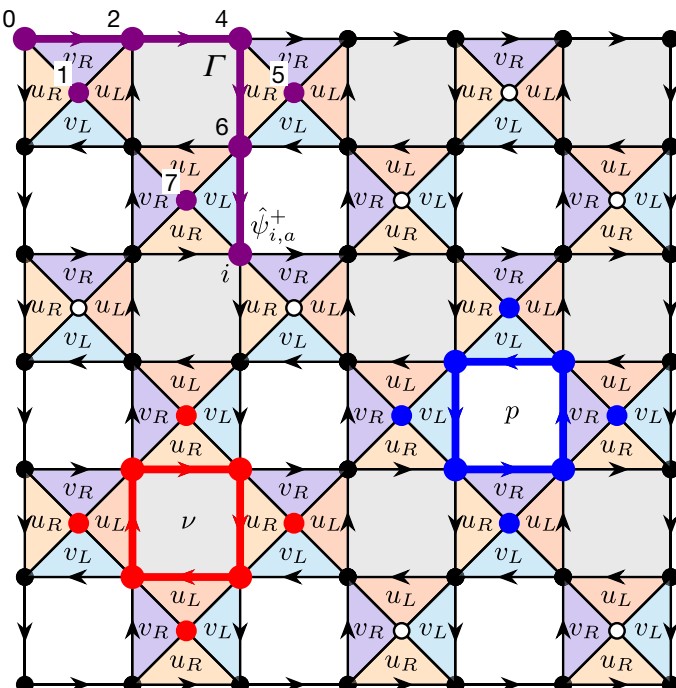

**Extended Data Fig. 1 | The 2D exactly solvable spin model on a 7 × 7 lattice with open boundary conditions.** Each black dot represents a 16-dimensional qudit on which the local operators $\hat{x}_{i,a}^{\pm}$ and $\hat{y}_{i,a}^{\pm}$ act and each open circle represents a 64-dimensional auxiliary qudit on which the local operators $\hat{w}_{ab}^{\pm}$ act, for $w = u_L$, $u_R$, $v_L$ or $v_R$. Each coloured triangle represents a three-body interaction between qudits on its three vertices. Also, we have eight-body interactions around every even plaquette (that is, the white and grey plaquettes). Equation (34) gives an example of a paraparticle operator $\hat{\psi}_{i,a}^{\pm}$ defined on the string $\Gamma$ (shown in purple), which is a MPO acting consecutively on all of the purple dots.

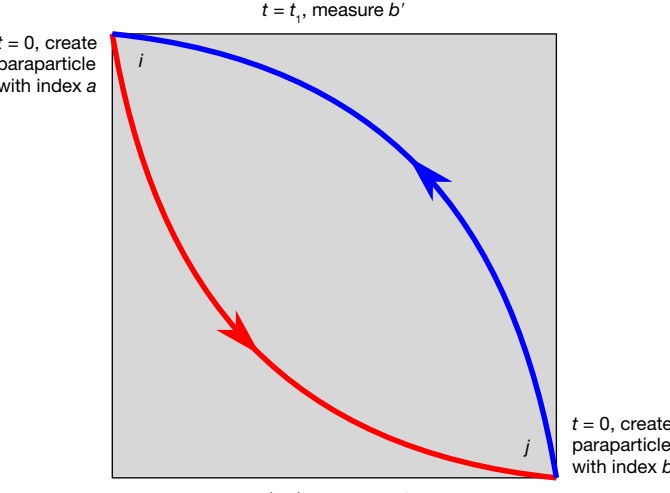

$t = 0$, create paraparticle with index $a$

$t = t_1$, measure $b'$

$t = 0$, create paraparticle with index $b$

$t = t_1$, measure $a'$

**Extended Data Fig. 2 | Illustration of paraparticle exchange in the 2D solvable spin model.** The shaded square represents the 2D system with open boundary conditions as shown in Extended Data Fig. 1. $i$ and $j$ label the black sites in the upper-left and lower-right corners of the 2D lattice, respectively, in which paraparticles can be locally created and measured. The unitary exchange operator $\hat{E}_{ij}$ moves the paraparticles along the two coloured paths and the result of the exchange is given in equation (13).