## [Peer Review File · Nature]

Particle exchange statistics beyond fermions and bosons

Corresponding Author: Dr Zhiyuan Wang

Version 0:

Reviewer comments:

Referee #1

(Remarks to the Author)

In this work, the authors propose a new construction for particles with parastatistics (that is: with more general exchange statistics than bosons or fermions), and demonstrate how these could emerge as quasiparticle excitations from physical condensed matter Hamiltonians.

Regarding the possibility for particles with parastatistics, a famous no-go theorem by Doplicher, Haag and Roberts (ref. [6]) states, under some basic locality assumptions, that those should be undistinguishable from ordinary fermions or bosons. The present work achieves a construction which escapes the conditions of this no-go theorem, as the paraparticles emerge as non-local objects from a local Hamiltonian, involving Jordan-Wigner strings or a generalization thereof.

The essential ingredient of the construction is the algebra satisfied by the paraparticle creation and annihilation operators, equation (6). The latter involves a so-called R matrix, satisfying some consistency conditions (5) reminiscent of the Yang-Baxter equation and the representation theory of braid groups, and which in some particular cases recovers the usual fermions and bosons. More generally, the properties of the R matrix allows to define a basis for the state space, defined in terms of a set of modes and the occupation numbers for each of these modes. A probe for the parastatistics is given by the thermal partition function in the case where the modes do not interact with one another, which in several given examples differs from that obtained in the cases of ordinary bosons or fermions. The authors then show how such non-interacting paraparticles may emerge from one-dimensional spin Hamiltonians, through a generalization of the Jordan-Wigner transformation.

This work is undeniably intriguing and original, providing a clear and persuasive account of how parastatistics emerge in one-dimensional systems. However, it's important to note that low-dimensional systems are already known to exhibit various exotic statistics, including parafermions and anyons. To bolster the impact of these findings, it is crucial for the authors to elucidate how their results can be extended to higher-dimensional systems, as they suggest. The current evidence for this extension is somewhat lacking.

In all the examples presented by the authors, the algebra (6) governing the quasiparticle creation and annihilation operations is contingent on the "R matrix," which is intimately linked to one-dimensional (or 1+1) systems. Therefore, the feasibility of these structures in higher-dimensional scenarios remains unclear. To enhance the paper, it would be invaluable if the authors could provide a concrete example of a higher-dimensional quantum spin Hamiltonian whose quasiparticle excitations exhibit non-trivial parastatistics. Without such an example, these results may be perceived as an intriguing construction in the realm of 1D physics, deserving publication in a more specialized journal.

Additional comments:

- Given the paper's emphasis on 1D systems, it would be beneficial for the authors to expound upon the distinctions between parastatistics and other types of statistics (parafermions, anyons) found in different systems. Do any of these statistics align with the framework presented here? While Comment [22] touches on this, more detailed insights would be enlightening.

- The authors consider thermal partition functions as a tool to probe parastatistics. However, as mentioned towards the end of page 5, the Hamiltonians they construct introduce superselection rules, partitioning the state space into sectors with fixed particle numbers, wherein the thermal partition functions resemble those of pure fermions or bosons. To truly establish parastatistics, the authors propose adding an infinitesimal amount of interaction to induce transitions between particle

number sectors. This suggestion is somewhat less convincing, as introducing interactions disrupts the non-interacting nature of the theory, making it challenging to discern statistics from the partition function. Can the authors confirm this point? Is there a way to modify the Hamiltonians (17) by introducing a term that alters the particle number while preserving the non-interacting structure of the theory?

In conclusion, this is an interesting work, but which does not yet demonstrate the level of generality announced by the authors. In the present form, I cannot recommend it for publication in Nature.

Referee #2

(Remarks to the Author)

In this paper, the authors discuss the existence of parastatistics which is not reduced to bosonic or fermionic statistics, and give explicit constructions especially in one dimension. The particle is assumed to have internal degrees of freedom (pseudospin), so that the R-matrix representing a two-particle exchange is nontrivial. While the parastatistics has been studied for many years, it is generally believed to be reduced to bosonic or fermionic statistics after all. In particular, Doplicher-Haag-Roberts "no-go theorem" implies the absence of nontrivial parastatistics based on the assumption of superselection rule. Despite these results, the authors argue that various kinds of nontrivial parastatistics in fact exist. Naturally, if the authors' claim indeed holds, it would be quite important.

Following the principle "extraordinary claims require extraordinary evidence", I tried to examine the paper critically. While I am not an expert on the "no-go theorem", I followed up the analysis of the spin model defined in Eq. (17) at least for the case "Ex. 3" in Table I. I verified that its energy spectrum is given in terms of "free paraparticles" as argued by the authors. As discussed in the paper, this implies that, although the partition function of the many-body system is given by a product of "single-mode partition function" $Z_R(x)$, it cannot be expressed in terms of bosons or fermions. This was enough to convince me that the authors' argument should be valid. I also find the discussion how their construction "evades" the no-go theorem reasonable.

Given these observations, I recommend publication of the present paper in Nature. However, I would like to ask the authors to answer the following points.

- The authors claim "although our spin model realization works only in 1D, our general formulation of paraparticles is valid in any dimension". However, I do not feel that the existence of paraparticles in higher dimensions is established in this paper. The authors should either present more concrete and convincing arguments for higher dimensions, or limit the main claim to the one dimension and leave the higher-dimensional generalization as a conjecture. (The one-dimensional construction is interesting enough so the second option is OK at least for me.)
- I suppose that the one-dimensional spin models solvable by the generalized Jordan-Wigner transformation to free paraparticles are a previously unknown class of integrable systems. Am I right? Or are some of them actually known integrable models and you are giving a new interpretation? Please clarify.

Referee #3

(Remarks to the Author)

This is an interesting manuscript, proposing a new setting for a notion of parastatistics in quantum mechanics. It relies on nontrivial representations of the symmetric group S_n , which are realized via R-matrices satisfying the Yang-Baxter equation (YBE).

The authors present a family of quantum spin models in 1D that realize their notion of parastatistics. They also observe that their scheme leads to a generalized exclusion principle (eq (13) and table 1). They carefully discuss how their realization avoids the no-go theorem of Doplicher-Haag-Roberts and discuss implications for realizing parastatistics for elementary particles (as opposed to excitations in a many-body system).

I find these results elegant and remarkable and worthy of publication in a first-rate journal. With this, I do have the following comments, which I'd like the authors to consider.

The authors rightfully point out that their scheme has similarities to related notions in the literature, such as Green's parastatistics, parafermions, non-Abelian anyons and fractional exclusion statistics. They defer their insights on this mostly to footnotes. To my taste these discussions are too concise and in places incomplete.

This holds in particular for the generalized exclusion principle. As early as 1940, Gentile [Nuovo Cimento 17: 493 (1940)] proposed such a notion, which in the notation of this paper would be the case $d_0=d_1=\dots d_l=1$ for any $l>0$. It is true that Haldane's fractional exclusion statistics is not defined in terms of algebraic properties in second quantization, but it has been recognized Haldane's statistics and generalizations thereof naturally appear in many-body partition functions in quantum field theory (CFT in particular), where they are closely related to algebraic properties of the (conformal) quantum fields. See for example [Phys.Rev.Lett.79:2608 (1997), Nucl.Phys.B547:501 (1999)]. I think these connections should be mentioned in this manuscript.

I also find the footnote [9] on the relation with non-Abelian anyons puzzling – given that there have been many systematic studies of non-Abelian anyons in two spatial dimensions, I would expect that a more precise statement on the relation with the current scheme can be made?

Referee #4

(Remarks to the Author)

Report on “Free particles beyond fermions and bosons”, Oct. 2nd, 2023.
by Z. Wang and K. R. A. Hazzard

The following points are itemized in accordance with Nature’s request:

%%%%%%%%%

A. Summary of the key results

The authors discuss that nontrivial parastatistics inequivalent to bosons and fermions can exist besides the two-dimensional anyonic statistics.

To prove this statement they constructed a series of models whose paraparticles admit internal indices. The braiding among paraparticles is given by Yang-Baxter type of matrices (contrary to quantum groups employed in integrable systems, they square to the identity, so that the paraparticles are a representation of the permutation group instead of the more general braid group).

They are able to compute the spectrum of the models and show, see e.g. figure 1, that, depending on the choice of the Yang-Baxter type of matrices, it cannot be reproduced by ordinary bosons/fermions. The presented models are non-interacting spin chains where the notion of locality is present. They further discuss a mechanism in which emergent paraparticles can evade the Doplicher-Haag-Roberts no-go theorem due to the fact that the quasiparticles in their spin models are created by nonlocal string operators.

%%%%%%%%%

B. Originality and significance: if not novel, please include reference

The paper is very interesting and significant. It focuses on the open question of the physical relevance of paraparticles. This question is prompted by recent experimental advances in condensed matter and more generally in engineering paraparticles in the laboratory (see below). The assumed naive extension of the DHR no-go theorem beyond its range of applicability has seriously hampered for many years the development of certain disciplines (I give an example below).

The authors use a second quantization method which encodes a notion of locality and are able to prove how to evade the hypotheses leading to the DHR no-go theorem. It should be stressed that the DHR no-go theorem can be evaded in a first-quantized formulation when no notion of locality is introduced.

The noninteracting theory that the authors consider as described by the type 2 Yang-Baxter matrix in Table 1 admits a first-quantized formulation and belongs to a class of theories investigated in both physics and mathematics (the Rittenberg-Wyler Z_2^n -graded Lie (super)algebras, the references are given below).

The algebra of the creation operators ψ^+_a, ψ^+_b for the $m=2$ value (then $a,b=1,2$; for simplicity the i,j indices are dropped) corresponds to vanishing commutators and anticommutators. The spinors are organized into two classes ($a,b=$ either 1 or 2). Spinors belonging to the $a=1$ class anticommute among themselves and satisfy the Pauli exclusion principle (similarly, spinors with $a=2$ anticommute), while $a=1$ spinors commute with $a=2$ spinors, generating a 2-bit parafermionic statistics, with particles organized as 00 (ordinary bosons), 10 (parafermionic spinors of type $a=1$), 01 (parafermionic spinors of type $a=2$), 11 “exotic” bosons.

Despite the parastatistics based on these (super)algebras having being investigated by decades, the formal proof that in a multiparticle sector the $Z_2 \times Z_2$ paraparticles are formally detectable has been given only recently in two papers in J. Phys. A, see the references below.

Applied to certain multiparticle states, an observable can spot the difference if the system is composed by $Z_2 \times Z_2$ paraparticles or by ordinary bosons/fermions.

The framework, following Majid’s, is that of a graded Hopf algebra endowed with a braided tensor product.

For too many years this field of investigation, despite being a true generalization of bosons/fermions, was dismissed by the misconception that Z_2^n -graded paraparticles do not lead to new results with respect to bosons/fermions (this situation started to change in the last 5 years).

The results presented by the authors in the submitted paper are in this line of research and are an extension of this activity.

%%%%%%%%%

C. Data & methodology: validity of approach, quality of data, quality of presentation

The presentation is accurate. The paper is very clearly written and readable (further points have been included in my other itemized remarks). I have spotted only one typo ("starightforwardly", at page 5 after citation [36]).

%%%%%%%%%

D. Appropriate use of statistics and treatment of uncertainties

The first part of the paper presents a construction which is mathematically treated with exact results. In the case of interacting paraparticles it is mentioned that standard approximation techniques can be employed.

%%%%%%%%%

E. The conclusions are correct and robust. The main statements are valid. The following statements must be corrected because they are misleading:

- it should be stressed that first-quantized theories can also evade the DHR no-go theorem because the notion of locality is not necessarily applicable in this setting.

- end of page 2, last paragraph:

It is incorrect to state that (in the case of fermions) the requirement of the physical observables to be composed by even products of fermionic operators is due to locality: the requirement is more fundamental than that and can be applied to quantum theories having no notion of locality. It derives from the fact that an observable applied to an eigenstate should produce a real number, not some fancy graded parameter like a Grassmann number.

- it should be stressed that the Yang-Baxter type of matrices introduced in the paper, contrary to the quantum groups Yang-Baxter matrices employed in integrable systems, square to the identity and give a representation of the permutation group.

%%%%%%%%%

F. Suggested improvements: experiments, data for possible revision

As mentioned in item B above, a first-quantized formalism for paraparticles is available as described in Chapter 10 of
- S. Majid,
Foundations of Quantum Group Theory
Cambridge University Press (1995).

The coproduct defined for a graded Hopf algebra endowed with a braided tensor product allows to construct multiparticle states. The connection between this formalism and the traditional Green's formulation of parastatistics based on the trilinear relations is discussed in

- B. Aneva and T. Popov,
Hopf Structure and Green Ansatz of Deformed Parastatistics Algebras,
J. Phys. A: Math. Gen. 38, 6473 (2005) (
{arXiv:math-ph/0412016}).
and

- K. Kanakoglou and C. Daskaloyannis,
Parabosons quotients. A braided look at Green's ansatz and a generalization,
J. Math. Phys. 48, 113516 (2007)
(arXiv:0901.04320).

I think the authors should mention these works since, as mentioned in point B, there seems to be a natural first-quantized formulation of the models introduced by the authors (this is definitely true for the R-matrices of type 2 entering Table I).

Concerning the R-matrices of type 3 in table I, I think the authors should make an effort in trying to identify them in the mathematical literature. These R-matrices seem to be quite a natural structure and could be possibly encountered. I didn't have time yet to make this check, but I am planning to consult the mathematical literature in this respect.

Furthermore, there are some interesting experimentalists' advances in the realm of simulation and engineering paraparticles in the laboratory; I think they deserved to be mentioned, see e.g.:

- C. Huerta Alderete et al.
Quantum simulation of driven para-Bose oscillators,
Phys. Rev. A 95, 013820 (2017)
(arXiv:1609.09166).

- C. Huerta Alderete et al.,
Experimental realization of para-particle oscillators,
(arXiv:2108.05471).

%%%%%%%%%

G. References: appropriate credit to previous work?

The reference [22] should be replaced to give credit to the recent results concerning the detectability of paraparticles from the Z_2^n -graded Lie algebras and superalgebras generating n-bit parastatistics and linked to the type 2 Yang-Baxter matrices of TABLE I.

These superalgebras
have been introduced by Rittenberg-Wyler in

- V. Rittenberg and D. Wyler,
Generalized Superalgebras
Nucl. Phys. B 139, 189 (1978).

- V.Rittenberg and D. Wyler,
Sequences of $Z_2 \times Z_2$ graded Lie algebras and superalgebras
J. Math. Phys. 19, 2193 (1978).

The proof that the $Z_2 \times Z_2$ paraparticles are theoretically detectable in a class of first-quantized quantum models has been recently presented in

- F. Toppan
 $Z_2 \times Z_2$ -graded parastatistics in multiparticle quantum Hamiltonians,
J. Phys. A: Math. Theor. 54, 115203 (2021)
(arXiv:2008.11554)

- F. Toppan
Inequivalent quantizations from gradings and $Z_2 \times Z_2$ parabosons,
J. Phys. A: Math. Theor. 54 (2021), 355202
(arXiv:2104.09692)

%%%%%%%%%

H. Clarity and context: lucidity of abstract/summary, appropriateness of abstract, introduction and conclusions

I have already praised the paper for clarity. Concerning other points, see my previous comments above.

%%%%%%%%%

Francesco Toppan
Full Professor CBPF - Theor. Dep., Rio de Janeiro (Brazil)
E-mail: toppan@cbpf.br

Version 1:

Reviewer comments:

Referee #1

(Remarks to the Author)

This revised version clarifies the main questions raised in my previous report. In particular, the authors now present an explicit 2D quantum Hamiltonian showing parastatistics, hence discarding doubts on the possibility to use their construction beyond one-dimensional systems.

A discussion on the difference between parastatistics and non-abelian anyons or parafermions in 2D was also presented in the Supplementary Material, which I find very enlightening. In particular, the authors stress that for anyons or parafermions the R-matrix does not square to identity, which forbids their existence in dimension higher than two. This is a basic fact, but quite crucial in my opinion, which I think deserves to be recalled in the main text.

Besides this minor suggestion, I now recommend this work for publication in Nature, and thank the authors for their thorough revision.

Referee #3

(Remarks to the Author)

I am satisfied with the changes made in the revised manuscript. In particular I appreciate the new section S1, detailing a comparison with different notions of fractional or para statistics that have featured in the literature. I also appreciate that the authors have now convincingly shown that their constructions are not restricted to one spatial dimension. With this I'm happy to recommend that the manuscript be published in Nature.

Referee #4

(Remarks to the Author)

The paper presents a class of solvable quantum spin models with emergent free paraparticles.

The authors are able to prove that the corresponding parastatistics cannot be reproduced by ordinary bosons/fermions statistics.

In the revised version the authors gave a detailed answer to the queries which were raised in the previous version.

Furthermore, in their reply to a Referee 1's question, they constructed a two-dimensional solvable quantum spin Hamiltonian with free emergent paraparticles excitations.

As already mentioned in my comments for the previous version, the paper is very interesting and significant. In my opinion the present version can be published as is.

Dear Editor,

Thank you for considering our manuscript “Free particles beyond fermions and bosons” for publication in Nature. We appreciate the Referees’ clear understanding of the results and positive assessments, as well as their useful comments.

Referees 2,3, and 4 have already recommended publication: Referee 2 finds our results “quite important”, Referee 3 finds our results “elegant and remarkable”, and Referee 4 finds our paper very interesting and significant. Referee 1 also finds our work “undeniably intriguing and original” but questioned the feasibility of extending our solvable spin models with emergent paraparticles to higher spatial dimensions and asked us to provide a concrete example of such a generalization. This is an important yet challenging research problem.

Our research in preparing the response to these Referee reports has fully settled this question, as we have now constructed a concrete example of a 2D solvable quantum spin Hamiltonian with free emergent paraparticle excitations. Our further analysis of the particle exchange statistics of these paraparticle excitations in higher dimensional systems reveals a striking physical difference from ordinary fermions and bosons. For this reason, we opened a new section in our paper for this analysis and changed our title to “Particle exchange statistics beyond fermions and bosons” to emphasize these new findings. The key definitions and main results of the 2D solvable model are described in Methods, and technical details are provided in the Supplementary Information.

We apologize for the delayed response – addressing this question required substantial new developments, and we felt that fully addressing this issue raised by Referee 1 (and mentioned by others) was important.

We have also edited our paper to address the other comments and suggestions of all the Referees, and our reply to each of the Referees’ point is given below. We strongly believe that the revised manuscript fully addresses the Referees’ concerns and meets Nature’s high standards for significance and broad impact, and we would like to resubmit our manuscript for your consideration.

Sincerely,
Zhiyuan Wang and Kaden R.A. Hazzard

Referees' comments:

Referee #1 (Remarks to the Author):

In this work, the authors propose a new construction for particles with parastatistics

(that is: with more general exchange statistics than bosons or fermions), and demonstrate how these could emerge as quasiparticle excitations from physical condensed matter Hamiltonians.

Regarding the possibility for particles with parastatistics, a famous no-go theorem by Doplicher, Haag and Roberts (ref. [6]) states, under some basic locality assumptions, that those should be undistinguishable from ordinary fermions or bosons. The present work achieves a construction which escapes the conditions of this no-go theorem, as the paraparticles emerge as non-local objects from a local Hamiltonian, involving Jordan-Wigner strings or a generalization thereof.

The essential ingredient of the construction is the algebra satisfied by the paraparticle creation and annihilation operators, equation (6). The latter involves a so-called R matrix, satisfying some consistency conditions (5) reminiscent of the Yang-Baxter equation and the representation theory of braid groups, and which in some particular cases recovers the usual fermions and bosons. More generally, the properties of the R matrix allows to define a basis for the state space, defined in terms of a set of modes and the occupation numbers for each of these modes. A probe for the parastatistics is given by the thermal partition function in the case where the modes do not interact with one another, which in several given examples differs from that obtained in the cases of ordinary bosons or fermions. The authors then show how such non-interacting paraparticles may emerge from one-dimensional spin Hamiltonians, through a generalization of the Jordan-Wigner transformation.

This work is undeniably intriguing and original, providing a clear and persuasive account of how parastatistics emerge in one-dimensional systems.

We thank the Referee for the accurate summary of our results and the positive assessments.

However, it's important to note that low-dimensional systems are already known to exhibit various exotic statistics, including parafermions and anyons. To bolster the impact of these findings, it is crucial for the authors to elucidate how their results can be extended to higher-dimensional systems, as they suggest. The current evidence for this extension is somewhat lacking.

In all the examples presented by the authors, the algebra (6) governing the quasiparticle creation and annihilation operations is contingent on the "R matrix," which is intimately linked to one-dimensional (or 1+1) systems. Therefore, the feasibility of these structures in higher-dimensional scenarios remains unclear. To enhance the paper, it would be invaluable if the authors could provide a concrete example of a higher-dimensional quantum spin Hamiltonian whose quasiparticle excitations exhibit non-trivial

parastatistics. Without such an example, these results may be perceived as an intriguing construction in the realm of 1D physics, deserving publication in a more specialized journal.

We thank the Referee for pointing out the limitation of our previous manuscript, and for highlighting that it wasn't clear that the the solvable spin models could be generalized to higher spatial dimensions.

Our revised manuscript has fully settled this issue: in Methods we describe a concrete example of a 2D solvable quantum spin Hamiltonian with emergent free paraparticle excitations, with additional technical details given in the Supplementary Information. Our analysis of the particle exchange statistics of these emergent paraparticles in higher dimensional systems reveals a striking physical difference from ordinary fermions and bosons (see the end of Methods). For this reason, we opened a new section in our paper for this analysis and changed our title to "Particle exchange statistics beyond fermions and bosons" to emphasize these new findings. Furthermore, as we discuss in Discussions and Methods, the new solvable spin model can host novel chiral and gapless topological phases of matter (depending on the free paraparticle spectrum) that are hard to study by previous techniques.

We expect that the techniques we used to construct and solve the 2D model can be generalized to construct 3D solvable spin models hosting emergent paraparticle excitations. Indeed, our 2D solvable spin model can be directly generalized to a large family of directed graphs (including a class of 3D lattices) where every vertex has exactly two incoming and two outgoing edges, and as long as the system has a unique ground state with a suitable open boundary condition, then we expect that most of our key results hold. For the 2D model shown in Fig.2 of this paper, we know the ground state is unique, using existing knowledge about Kitaev's quantum double models (based on Hopf algebras) in 2D. An additional technical challenge in 3D is that Kitaev's quantum double models (based on Hopf algebras) have not yet been generalized to 3D, so we may need some different technical tools to study the ground state properties of the generalizations of our models in 3D. We leave this to a future work. As they stand, the current results clearly show that the ideas are not associated with 1D physics, can be generalized to 2D, and no apparent fundamental obstacles to constructing 3D models stand in the way.

Additional comments:

- Given the paper's emphasis on 1D systems, it would be beneficial for the authors to expound upon the distinctions between parastatistics and other types of statistics (parafermions, anyons) found in different systems. Do any of these statistics align with the framework presented here? While Comment [22] touches on this, more detailed

insights would be enlightening.

We thank the Referee for this comment. We have written a new section S1 in the Supplementary Information where we explain the difference between parastatistics and other types of particle statistics, including non-Abelian anyons, parafermions, and other types of exclusion statistics. We also explain the difference between our theory of parastatistics and Green's theory.

- The authors consider thermal partition functions as a tool to probe parastatistics. However, as mentioned towards the end of page 5, the Hamiltonians they construct introduce superselection rules, partitioning the state space into sectors with fixed particle numbers, wherein the thermal partition functions resemble those of pure fermions or bosons. To truly establish parastatistics, the authors propose adding an infinitesimal amount of interaction to induce transitions between particle number sectors. This suggestion is somewhat less convincing, as introducing interactions disrupts the non-interacting nature of the theory, making it challenging to discern statistics from the partition function. Can the authors confirm this point? Is there a way to modify the Hamiltonians (17) by introducing a term that alters the particle number while preserving the non-interacting structure of the theory?

We thank the Referee for this question. We first would like to note that in thermodynamics and statistical mechanics, what people mean by “free particle systems” or “non-interacting systems” is actually a weakly interacting system in the limit of zero-interaction. The reason is that a free particle system with strictly zero interaction will never thermalize, so we cannot expect the laws of equilibrium statistical mechanics (e.g. the ideal gas law) to apply in such a system. The way we use infinitesimal perturbation by spin-flip operators to thermalize the system is analogous to how infinitesimal interaction thermalizes an ideal gas. One can make this precise by performing perturbation theory in the interaction term. This shows that at all non-zero temperatures, the observables are unaffected by an infinitesimal interaction term.

Alternatively, we can imagine a quantum dynamical process in which we completely shutdown the spin-flip perturbations after the system fully thermalizes, so that the perturbation will not affect the non-interacting nature of the resulting free paraparticle system in thermal equilibrium, as the spin-flip perturbation is only used in the non-equilibrium dynamical process for the system to fully thermalize.

Regarding the Referee's question about number-non-conservation, we point out that the total particle number is not the only conserved observable that commutes with the local Hamiltonian terms, and each particle number sector further breaks into a direct sum of subspaces (superselection sectors) such that the local Hamiltonian terms (or more generally, any observable that is local in the paraparticle picture) drive no transition

between different superselection sectors. We can add local pair creation and annihilation terms into the spin Hamiltonian Eq. (17) or the free pararticle Hamiltonian (19), analogous to Bogoliubov's mean-field theory of superconductors, which breaks the total particle number but still preserves the solvability of free pararticles. However, this still does not fully break all the superselection rules. In summary, we do need the spin-flip terms to break all the superselection rules and allow full thermalization to the pararticle partition function.

In conclusion, this is an interesting work, but which does not yet demonstrate the level of generality announced by the authors. In the present form, I cannot recommend it for publication in Nature.

We would like to thank the Referee again for the constructive comments and criticisms. Now that the Referee's main concern about extending the spin model to higher dimensions is solved, we would like the Referee to reconsider the significance and potential impact of our work.

Referee #2 (Remarks to the Author):

In this paper, the authors discuss the existence of parastatistics which is not reduced to bosonic or fermionic statistics, and give explicit constructions especially in one dimension. The particle is assumed to have internal degrees of freedom (pseudospin), so that the R-matrix representing a two-particle exchange is nontrivial. While the parastatistics has been studied for many years, it is generally believed to be reduced to bosonic or fermionic statistics after all. In particular, Doplicher-Haag-Roberts "no-go theorem" implies the absence of nontrivial parastatistics based on the assumption of superselection rule. Despite these results, the authors argue that various kinds of nontrivial parastatistics in fact exist. Naturally, if the authors' claim indeed holds, it would be quite important.

Following the principle "extraordinary claims require extraordinary evidence", I tried to examine the paper critically. While I am not an expert on the "no-go theorem", I followed up the analysis of the spin model defined in Eq. (17) at least for the case "Ex. 3" in Table I. I verified that its energy spectrum is given in terms of "free pararticles" as argued by the authors. As discussed in the paper, this implies that, although the partition function of the many-body system is given by a product of "single-mode partition function" $Z_R(x)$, it cannot be expressed in terms of bosons or fermions. This was enough to convince me that the authors' argument should be valid. I also find the discussion how their construction "evades" the no-go theorem reasonable.

Given these observations, I recommend publication of the present paper in Nature.

We thank the Referee for checking the arguments, their summary, and the positive assessment of our work.

However, I would like to ask the authors to answer the following points.

- The authors claim "although our spin model realization works only in 1D, our general formulation of paraparticles is valid in any dimension". However, I do not feel that the existence of paraparticles in higher dimensions is established in this paper. The authors should either present more concrete and convincing arguments for higher dimensions, or limit the main claim to the one dimension and leave the higher-dimensional generalization as a conjecture. (The one-dimensional construction is interesting enough so the second option is OK at least for me.)

We thank the Referee for raising this concern, echoing Referee 1's concern. Our revised manuscript has fully addressed it: we have constructed a 2D solvable spin model hosting emergent free paraparticles, whose exchange statistics shows a striking physical difference from ordinary fermions and bosons. (For this reason we changed our title to "Particle exchange statistics beyond fermions and bosons" to emphasize their distinct exchange statistics.) It is discussed in the main text, the construction is given in Methods, and detailed definitions and proofs are given in the Supplementary Information. See also our reply to Referee #1.

- I suppose that the one-dimensional spin models solvable by the generalized Jordan-Wigner transformation to free paraparticles are a previously unknown class of integrable systems. Am I right? Or are some of them actually known integrable models and you are giving a new interpretation? Please clarify.

The construction of the 1D solvable spin models with emergent free paraparticles based on an arbitrary R-matrix is completely new. However, for some special choice of the R-matrix, the model is known, or is equivalent to some known models. For example, for the R-matrix in Ex.1 and Ex.2 of Table I, the model belongs to the class of free-fermion solvable models that has already been classified. The model based on Ex.3 has appeared in a previous publication (coauthored by us) PRA 99, 013624 (2019) without realizing its relation to parastatistics, and the model was solved by an alternative method. Those are the only three known cases, to the best of our knowledge. In particular, the model based on the R-matrix in Ex.4 of Table I and the set-theoretical R-matrix in Eqs. (34,35) are new.

The 2D solvable spin model we added in the new edition is completely new and has qualitatively different and novel physical phenomena compared to previous solvable models, see our discussion in the main text below Eq. (19) and in Methods.

Referee #3 (Remarks to the Author):

This is an interesting manuscript, proposing a new setting for a notion of parastatistics in quantum mechanics. It relies on nontrivial representations of the symmetric group S_n , which are realized via R-matrices satisfying the Yang-Baxter equation (YBE).

The authors present a family of quantum spin models in 1D that realize their notion of parastatistics. They also observe that their scheme leads to a generalized exclusion principle (eq (13) and table 1). They carefully discuss how their realization avoids the no-go theorem of Doplicher-Haag-Roberts and discuss implications for realizing parastatistics for elementary particles (as opposed to excitations in a many-body system).

I find these results elegant and remarkable and worthy of publication in a first-rate journal.

We appreciate and agree with the summary and thank the Referee for their appreciation of the results.

With this, I do have the following comments, which I'd like the authors to consider.

The authors rightfully point out that their scheme has similarities to related notions in the literature, such as Green's parastatistics, parafermions, non-Abelian anyons and fractional exclusion statistics. They defer their insights on this mostly to footnotes. To my taste these discussions are too concise and in places incomplete.

This holds in particular for the generalized exclusion principle. As early as 1940, Gentile [Nuovo Cimento 17: 493 (1940)] proposed such a notion, which in the notation of this paper would be the case $d_0=d_1=\dots d_l=1$ for any $l>0$. It is true that Haldane's fractional exclusion statistics is not defined in terms of algebraic properties in second quantization, but it has been recognized Haldane's statistics and generalizations thereof naturally appear in many-body partition functions in quantum field theory (CFT in particular), where they are closely related to algebraic properties of the (conformal) quantum fields. See for example [Phys.Rev.Lett.79:2608 (1997), Nucl.Phys.B547:501 (1999)]. I think these connections should be mentioned in this manuscript.

I also find the footnote [9] on the relation with non-Abelian anyons puzzling – given that there have been many systematic studies of non-Abelian anyons in two spatial dimensions, I would expect that a more precise statement on the relation with the

current scheme can be made?

We thank the Referee for this suggestion and the references on generalized exclusion statistics. We have written a new section S1 in the Supplementary Information where we summarize the differences between parastatistics and other known types of particle statistics, including non-Abelian anyons, parafermions, and other types of exclusion statistics (including the one suggested by the Referee), and the difference between our theory of parastatistics and Green's theory.

Referee #4 (Remarks to the Author):

*Report on "Free particles beyond fermions and bosons", Oct. 2nd, 2023.
by Z. Wang and K. R. A. Hazzard*

The following points are itemized in accordance with Nature's request:

%%%%%%%%%

A. Summary of the key results

The authors discuss that nontrivial parastatistics inequivalent to bosons and fermions can exist besides the two-dimensional anyonic statistics.

To prove this statement they constructed a series of models whose paraparticles admit internal indices.

The braiding among paraparticles is given by Yang-Baxter type of matrices (contrary to quantum groups employed in integrable systems, they square to the identity, so that the paraparticles are a representation of the permutation group instead of the more general braid group).

They are able to compute the spectrum of the models and show, see e.g. figure 1, that, depending on the choice of the Yang-Baxter type of matrices, it cannot be reproduced by ordinary bosons/fermions.

The presented models are non-interacting spin chains where the notion of locality is present. They further discuss a mechanism in which emergent paraparticles can evade the Doplicher-Haag-Roberts no-go theorem due to the fact that the quasiparticles in their spin models are created by nonlocal string operators.

We thank the Referee for the nice summary of our results. We only want to make a small clarification of one phrase in the summary: our spin model Hamiltonians are actually strongly interacting, while they map to non-interacting paraparticles through the non-local MPO Jordan-Wigner transformation. (We suspect this is what the Referee intended

with the shorter phrase “non-interacting spin chain”, but wanted to explicitly note this to clarify)

%%%%%%%%%

B. Originality and significance: if not novel, please include reference

The paper is very interesting and significant. It focuses on the open question of the physical relevance of paraparticles. This question is prompted by recent experimental advances

in condensed matter and more generally in engineering paraparticles in the laboratory (see below). The assumed naive extension of the DHR no-go theorem beyond its range of applicability has seriously hampered for many years the development of certain disciplines (I give an example below).

The authors use a second quantization method which encodes a notion of locality and are able to prove how to evade the hypotheses leading to the DHR no-go theorem. It should be stressed that the DHR no-go theorem can be evaded in a first-quantized formulation when no notion of locality is introduced.

The noninteracting theory that the authors consider as described by the type 2 Yang-Baxter matrix in Table 1 admits a first-quantized formulation and belongs to a class of theories investigated in both physics and mathematics (the Rittenberg-Wyler Z_2^n -graded Lie (super)algebras, the references are given below).

The algebra of the creation

operators ψ^+_a, ψ^+_b for the $m=2$ value (then $a,b=1,2$; for simplicity the i,j indices are dropped) corresponds to vanishing commutators and anticommutators. The spinors are organized into two classes ($a,b=$ either 1 or 2). Spinors belonging to the $a=1$ class anticommute among themselves and satisfy the Pauli exclusion principle (similarly, spinors with $a=2$ anticommute), while $a=1$ spinors commute with $a=2$ spinors, generating a 2-bit parafermionic statistics, with particles organized as 00 (ordinary bosons), 10 (parafermionic spinors of type $a=1$), 01 (parafermionic spinors of type $a=2$), 11 “exotic” bosons.

Despite the parastatistics based on these (super)algebras having being investigated by decades, the formal proof that in a multiparticle sector the $Z_2 \times Z_2$ paraparticles are formally detectable has been

given only recently in two papers in J. Phys. A, see the references below.

Applied to certain multiparticle states, an observable can spot the difference if the system is composed by $Z_2 \times Z_2$ paraparticles or by ordinary bosons/fermions.

The framework, following Majid’s, is that of a graded Hopf algebra endowed with a

braided tensor product.

For too many years this field of investigation, despite being a true generalization of bosons/fermions, was dismissed by the misconception that Z_2^n -graded paraparticles do not lead to new results with respect to bosons/fermions (this situation started to change in the last 5 years).

The results presented by the authors in the submitted paper are in this line of research and are an extension of this activity.

We agree that there are many misconceptions about particle statistics that are fortunately being remedied, and we thank the Referee for giving us the background about the type-2 R-matrix in our Table I. We also appreciate the mathematical structure the Referee has mentioned, and we agree that this is an interesting research area.

%%%%%%%%%

C. Data & methodology: validity of approach, quality of data, quality of presentation

The presentation is accurate. The paper is very clearly written and readable (further points have been included in my other itemized remarks). I have spotted only one typo (“starightforwardly”, at page 5 after citation [36]).

We thank the Referee for the positive assessments. We have corrected this typo.

%%%%%%%%%

D. Appropriate use of statistics and treatment of uncertainties

The first part of the paper presents a construction which is mathematically treated with exact results. In the case of interacting paraparticles it is mentioned that standard approximation techniques can be employed.

%%%%%%%%%

E. The conclusions are correct and robust. The main statements are valid. The following statements must be corrected because they are misleading:

- it should be stressed that first-quantized theories can also evade the DHR no-go theorem because the notion of locality is not necessarily applicable in this setting.

We thank the Referee for pointing this out, and we now mention this in the footnote [21] in the revised manuscript.

- end of page 2, last paragraph:

It is incorrect to state that (in the case of fermions) the requirement of the physical observables to be composed by even products of fermionic operators is due to locality: the requirement is more fundamental than that and can be applied to quantum theories having no notion of locality. It derives from the fact that an observable applied to an eigenstate should produce a real number, not some fancy graded parameter like a Grassmann number.

We agree with the Referee that the requirement of observables having even fermion parity can be applied to quantum theories having no notion of locality: a famous example is the Sachdev-Ye-Kitaev model.

However, we would also like to point out that

(1). It is still not wrong to say that requiring local observables to have even fermion parity is necessary for a local quantum field theory, since having observables with odd fermion parity at two different spatial locations necessarily violates locality;
(2). Although the following idea may sound weird when considering fundamental particles, having a Hamiltonian with one single fermionic operator at a boundary point does not break locality, and it has some physical applications in condensed matter physics: under the Jordan-Wigner transformation, this maps to a locally-interacting quantum spin Hamiltonian with a spin-flip operator at the boundary, which is totally physical. We emphasize that in this case, the eigenvalues of the fermionic operator are still ordinary numbers, since we are using the second quantization formulation and there is no Grassmann number in the theory. Only after one extends the base field from complex numbers to the Grassmann numbers does one encounter fermion coherent states that are eigenstates of fermionic operators with eigenvalues being Grassmann numbers.

- it should be stressed that the Yang-Baxter type of matrices introduced in the paper, contrary to the quantum groups Yang-Baxter matrices employed in integrable systems, square to the identity and give a representation of the permutation group.

We thank the Referee for mentioning the connection to quantum groups [quasitriangular Hopf algebras (QHA)]. Indeed, the relation between the R-matrices in this paper and QHA is quite subtle.

First, notice that the R-matrices constructed from triangular Hopf algebras [THAs, which are QHAs in which the universal R-matrix \underline{R} satisfies the additional requirement $\underline{R}_{21}\underline{R}=I$] are guaranteed to be involutive, $R^2=I$, and satisfy the YBE. Indeed, some R-matrices that appeared in our manuscript can be constructed from finite dimensional

THAs, such as Ex.1 and Ex.2 in Table I, and the newly added set-theoretical R-matrix in Eqs. (34,35). Paraparticles based on this type of R-matrix are guaranteed to be realizable in 2D quantum spin systems, and our construction of the 2D solvable spin model is based on this underlying Hopf algebra structure.

The other two R-matrices in our paper, Ex.3 and Ex.4 in Table I, provably cannot be constructed from finite dimensional THAs, and their realizability in 2D quantum spin models is open.

The R-matrices employed in quantum integrable systems depend on two spectral parameters, and cannot be used to define parastatistics (to our knowledge).

We think that in our current manuscript, it is clear that all the R-matrices we consider satisfy the YBE, square to identity, and therefore define representation of S_n , as is emphasized below Eq. (4) and Eq. (5).

%%%%%%%%%

F. Suggested improvements: experiments, data for possible revision

As mentioned in item B above, a first-quantized formalism for paraparticles is available as described in Chapter 10 of

- S. Majid,

Foundations of Quantum Group Theory

Cambridge University Press (1995).

The coproduct defined for a graded Hopf algebra endowed with a braided tensor product allows to construct multiparticle states.

We don't fully understand the connection between the first quantization of paraparticles and Chapter 10 of Majid's book, but in the SI we cited that book as a reference for triangular Hopf algebras.

The connection between this formalism and the traditional Green's formulation of parastatistics based on the trilinear relations is discussed in

- B. Aneva and T. Popov,

Hopf Structure and Green Ansatz of Deformed Parastatistics Algebras,

J. Phys. A: Math. Gen. 38, 6473 (2005) (

{arXiv:math-ph/0412016}.

and

- K. Kanakoglou and C. Daskaloyannis,

Parabosons quotients. A braided look at Green's ansatz and a generalization, J. Math. Phys. 48, 113516 (2007) (arXiv:0901.04320).

We thank the Referee for mentioning these papers. We mentioned them in Sec. S1 of the revised SI.

I think the authors should mention these works since, as mentioned in point B, there seems to be a natural first-quantized formulation of the models introduced by the authors (this is definitely true for the R-matrices of type 2 entering Table I).

The Referee's intuition is correct: indeed, Sec. S3 in the SI is devoted to explaining the relation between the first and second quantization formulation in this paper.

Concerning the R-matrices of type 3 in table I, I think the authors should make an effort in trying to identify them in the mathematical literature. These R-matrices seem to be quite a natural structure and could be possibly encountered. I didn't have time yet to make this check, but I am planning to consult the mathematical literature in this respect.

The type-3 R-matrix in Table I is proportional to the identity matrix and is often used as a trivial example in introductory texts to the YBE [e.g. Examples in the Wikipedia page on Yang–Baxter equation: https://en.wikipedia.org/wiki/Yang%E2%80%93Baxter_equation#Set-theoretic_Yang%E2%80%93Baxter_equation]. The type-4 R-matrix in Table I appears to be new, to the best of our knowledge. Indeed, we asked a related question on this issue on the website <https://mathoverflow.net/questions/391667/involutive-solutions-to-the-yang-baxter-equation-and-triangular-hopf-algebras>, and R-matrices with Hilbert series inequivalent to the first three examples appear to be quite rare. Another thing we know about the type-3 and type-4 R-matrices is that they provably cannot be constructed from any finite dimensional (quasi)triangular Hopf algebras.

Furthermore, there are some interesting experimentalists' advances in the realm of simulation and engineering paraparticles in the laboratory; I think they deserved to be mentioned, see e.g.:

- C. Huerta Alderete et al.
Quantum simulation of driven para-Bose oscillators, Phys. Rev. A 95, 013820 (2017) (arXiv:1609.09166).

- C. Huerta Alderete et al.,

*Experimental realization of para-particle oscillators,
(arXiv:2108.05471).*

We thank the Referee for mentioning these experimental works realizing Green's parastatistics in small systems. We mention them in Sec. S1 of the revised SI when comparing Green's parastatistics to our construction.

%%%%%%%%%

G. References: appropriate credit to previous work?

The reference [22] should be replaced to give credit to the recent results concerning the detectability of paraparticles from the Z_2^n -graded Lie algebras and superalgebras generating n-bit parastatistics and linked to the type 2 Yang-Baxter matrices of TABLE I.

These superalgebras have been introduced by Rittenberg-Wyler in

*- V. Rittenberg and D. Wyler,
Generalized Superalgebras
Nucl. Phys. B 139, 189 (1978).*

*- V.Rittenberg and D. Wyler,
Sequences of $Z_2 \times Z_2$ graded Lie algebras and superalgebras
J. Math. Phys. 19, 2193 (1978).*

The proof that the $Z_2 \times Z_2$ paraparticles are theoretically detectable in a class of first-quantized quantum models has been recently presented in

*- F. Toppan
 $Z_2 \times Z_2$ -graded parastatistics in multiparticle quantum Hamiltonians,
J. Phys. A: Math. Theor. 54, 115203 (2021)
(arXiv:2008.11554)*

*- F. Toppan
Inequivalent quantizations from gradings and $Z_2 \times Z_2$ parabosons,
J. Phys. A: Math. Theor. 54 (2021), 355202
(arXiv:2104.09692)*

We thank the Referee for mentioning these works. We mention them in two places in the SI: in Sec.S1, where we clarify the relation and difference between our theory of parastatistics and Green's theory, and in a footnote in Sec. S5.A, where we mention the

possibility of realizing emergent paraparticles in higher dimensional fermionic systems, where (Hopf) superalgebras may be a useful tool.

%%%%%%%%%

H. Clarity and context: lucidity of abstract/summary, appropriateness of abstract, introduction and conclusions

I have already praised the paper for clarity. Concerning other points, see my previous comments above.

We thank the Referee again for the appreciation of our work and the constructive comments.

Dear Editor,

Thank you for considering our manuscript “Particle exchange statistics beyond fermions and bosons” for publication in Nature. We are happy to see that all the Referees appreciate our new result of the 2D solvable spin model added to the manuscript and that they all recommend publication in Nature. We also appreciate the additional suggestion from Referee #1 and we have implemented this suggestion in the updated manuscript (see our response below).

We have also shortened the main text of our paper to address the length concern and edited the format according to Nature’s formatting guide. In particular, we moved the discussion of superselection rules and DHR theorem (in the paragraph “Speculations about elementary paraparticles” at the end of the main text) to Sec. S6 of the SI, and moved a few sentences of the section “Exact solution of free paraparticles” to the corresponding section in Methods. In total, this shortens the paper by about 500 words.

We believe that the revised manuscript meets Nature’s length requirements and formatting rules, and we would like to resubmit our manuscript for acceptance.

Furthermore, we wish to participate in transparent peer review.

Sincerely,
Zhiyuan Wang and Kaden R.A. Hazzard

Referees' comments:

Referee #1 (Remarks to the Author):

This revised version clarifies the main questions raised in my previous report. In particular, the authors now present an explicit 2D quantum Hamiltonian showing parastatistics, hence discarding doubts on the possibility to use their construction beyond one-dimensional systems.

A discussion on the difference between parastatistics and non-abelian anyons or parafermions in 2D was also presented in the Supplementary Material, which I find very enlightening. In particular, the authors stress that for anyons or parafermions the R-matrix does not square to identity, which forbids their existence in dimension higher than two. This is a basic fact, but quite crucial in my opinion, which I think deserves to be recalled in the main text.

Besides this minor suggestion, I now recommend this work for publication in Nature, and thank the authors for their thorough revision.

We thank the Referee for the useful suggestion and the recommendation for publication. We implemented this suggestion by adding a sentence at the end of the second paragraph “Notice that the first relation in Eq. (4) is crucial for parastatistics to be consistently defined in any dimension; anyons generally do not satisfy this relation, and consequently they only form a representation of the braid group B_n [16] instead of the symmetric group S_n , and are therefore limited to 2D.”.

Referee #3 (Remarks to the Author):

I am satisfied with the changes made in the revised manuscript. In particular I appreciate the new section S1, detailing a comparison with different notions of fractional or para statistics that have featured in the literature. I also appreciate that the authors have now convincingly shown that their constructions are not restricted to one spatial dimension. With this I'm happy to recommend that the manuscript be published in Nature.

We would like to thank the Referee for the appreciation of our results and the recommendation for publication.

Referee #4 (Remarks to the Author):

The paper presents a class of solvable quantum spin models with emergent free paraparticles. The authors are able to prove that the corresponding parastatistics cannot be reproduced by ordinary bosons/fermions statistics. In the revised version the authors gave a detailed answer to the queries which were raised in the previous version. Furthermore, in their reply to a Referee 1's question, they constructed a two-dimensional solvable quantum spin Hamiltonian with free emergent paraparticles excitations.

As already mentioned in my comments for the previous version, the paper is very interesting and significant. In my opinion the present version can be published as is.

We would like to thank the Referee for the efforts in reviewing the first and the second version of our manuscript and the recommendation for publication.